# *Mycobacterium tuberculosis* resides in lysosome-poor monocyte-derived lung cells during chronic infection

Weihao Zheng[1], I-Chang Chang[1], Jason Limberis[1], Jonathan M. Budzik[2], Beth Shoshana Zha[1,2], Zachary Howard[1], Lucas Chen[1], Joel D. Ernst[1] *

1 Division of Experimental Medicine, Department of Medicine, University of California, San Francisco, California, United States of America, 2 Division of Pulmonary, Critical Care, Allergy and Sleep Medicine, Department of Medicine, University of California, San Francisco, California, United States of America

* joel.ernst@ucsf.edu

**Data Availability Statement:** RNA-seq data were deposited in GEO under accession number GSE220147.

## Abstract

*Mycobacterium tuberculosis* (Mtb) infects lung myeloid cells, but the specific Mtb-permissive cells and host mechanisms supporting Mtb persistence during chronic infection are incompletely characterized. We report that after the development of T cell responses, CD11c[lo] monocyte-derived cells harbor more live Mtb than alveolar macrophages (AM), neutrophils, and CD11c[hi] monocyte-derived cells. Transcriptomic and functional studies revealed that the lysosome pathway is underexpressed in this highly permissive subset, characterized by less lysosome content, acidification, and proteolytic activity than AM, along with less nuclear TFEB, a regulator of lysosome biogenesis. Mtb infection does not drive lysosome deficiency in CD11c[lo] monocyte-derived cells but promotes recruitment of monocytes that develop into permissive lung cells, mediated by the Mtb ESX-1 secretion system. The c-Abl tyrosine kinase inhibitor nilotinib activates TFEB and enhances lysosome functions of macrophages in vitro and in vivo, improving control of Mtb infection. Our results suggest that Mtb exploits lysosome-poor lung cells for persistence and targeting lysosome biogenesis is a potential host-directed therapy for tuberculosis.

## Author summary

*Mycobacterium tuberculosis* is notorious for its ability to replicate in mononuclear phagocytes and cause chronic lung infection despite development of immune responses. However, emerging evidence indicates that, depending on the stage of infection, some macrophages are more permissive than others for intracellular *M. tuberculosis* survival. We discovered that CD11c[lo] monocyte-derived cells harbor more live *M. tuberculosis* than other lung myeloid cells such as alveolar macrophages, neutrophils, and CD11c[hi] monocyte-derived cells when assessed after full development of adaptive immune responses. Compared to alveolar macrophages, these cells show reduced nuclear TFEB, a regulator of lysosome biogenesis, and lysosome activity, a crucial defense mechanism. *M. tuberculosis* infection doesn't directly induce lysosome deficiency in infected cells; instead, it recruits

**Funding:** U01 AI166309 (JDE) National Institute of Allergy and Infectious Diseases, R01 AI51242 (JDE) National Institute of Allergy and Infectious Diseases. nih.gov The funders did not play a role in the study design, data collection and analysis, decision to publish, or preparation of the manuscript.

**Competing interests:** The authors have declared that no competing interests exist.

monocytes that develop into permissive lung cells, facilitated by the Mtb ESX-1 secretion system. The c-Abl tyrosine kinase inhibitor nilotinib activates TFEB, enhancing lysosome functions in macrophages and improving the control of *M. tuberculosis* infection in vitro and in vivo. Our findings providing new insights into the host mechanisms involved in *M. tuberculosis* persistence, shedding new light on developing effective treatments for tuberculosis by targeting lysosome biogenesis and addressing the challenge of persistent infections.

## Introduction

A distinguishing characteristic of *Mycobacterium tuberculosis* (Mtb) is its ability to evade elimination by innate and adaptive immune responses, leading to chronic infection with lung granuloma formation as a hallmark [1]. The ability of Mtb to avoid elimination poses challenges to vaccine development [1,2] and facilitates its transmission, thereby contributing to the ongoing global pandemic of tuberculosis (TB) [3]. For these reasons, it is important to identify the permissive cellular niche of Mtb in vivo and determine the mechanisms that allow Mtb to persist in the face of innate and adaptive immunity.

Mtb is a facultative intracellular pathogen, and resides predominantly in mononuclear phagocytes, including resident tissue (i.e., alveolar) macrophages and in cells derived from circulating monocytes [4–12]. There is also substantial evidence that the fate of pathogenic mycobacteria in distinct cell types can differ in vivo during chronic infection. Nearly 100 years ago, Florence Sabin reported two distinct cell types in vivo that differed in their handling of pathogenic mycobacteria: 'clasmatocytes' (tissue resident macrophages) "...phagocytize tubercle bacilli freely and fragment them", while monocytes "retain the tubercle bacilli intact, with power to survive and multiply, over long periods of time" [11]. These findings indicate that distinct types of mononuclear cells differ in their capacity to control pathogenic mycobacteria, but the identity of the cells that harbor Mtb and the mechanisms that determine their differential abilities to control Mtb during chronic infection are incompletely understood.

Recent studies using Mtb strains that constitutively express fluorescent proteins have confirmed that alveolar macrophages (AM), the tissue-resident macrophages of the air spaces, are the initial targets of infection [5,7,10,13]. During the initial 7–14 days of infection, Mtb replicates efficiently in AM in vivo [5,7,8,10], and there is evidence that AM are less Mtb-restrictive than lung interstitial macrophages (IM) in the innate immune stage of infection [7]. However, the AM population is finite and does not expand markedly in response to infection [8,9,12]. Therefore, for Mtb to expand its population and maximize the likelihood of transmission, the bacteria spread beyond AM.

One of the responses to Mtb infection is the recruitment of inflammatory cells, especially monocytes and neutrophils, to the lungs [8,9,12,14,15]. Monocytes develop from progenitors in the bone marrow [16] before entering the bloodstream, a step that depends on the chemokine receptor, CCR2 [17]. In mice infected with Mtb, monocytes migrate from the blood to the lung parenchyma and differentiate into two distinct cell subsets distinguished by their expression of CD11c [9]. CD11c[hi] monocyte-derived cells were formerly considered dendritic cells [12,15,18], although they have also been considered closely related to macrophages based on their transcriptional profiles [8], while CD11c[lo] monocyte-derived cells have been termed recruited macrophages [12,15,19]. Both of these cell subsets become infected with Mtb within 3–5 days after they enter the lung parenchyma [9], and cells in both of these subsets increase in number and frequency in the lungs for at least 16 weeks post infection [9]. Using fluorescent

protein-expressing Mtb and the intensity of bacterial fluorescence per cell as a readout, a population of CD11c[hi] monocyte-derived lung cells has been reported to be infected with high frequency by Mtb [8].

One consequence of Mtb spread from AM to other cell types is the transport of live bacteria from the lungs to the local draining lymph nodes [18], where bacterial antigens are transferred from infected migratory dendritic cells to uninfected resident lymph node dendritic cells for antigen-specific T cell priming [20,21]. Upon arrival of CD4 effector T cells in the lungs, the Mtb population stabilizes but is not eliminated. Together, these observations suggest that the cells in which Mtb resides after the acute stage of infection (≥4 weeks) cannot kill the bacteria at a rate greater than their growth, despite the presence of effector T cells.

Considering the events described above, the course of the host response to Mtb infection can be considered in at least 3 distinct stages. In mice, the initial stage is the first 10–14 days of infection, before development of antigen-specific CD4 or CD8 T cell responses. During the latter portion of the initial stage, inflammatory cells including neutrophils and monocytes are recruited to the lungs [12,15]. The second stage is transitional, comprising approximately 15–25 days post infection (dpi), and is marked by an accumulation of monocytes and their differentiation in the lung parenchyma, together with the appearance of effector CD4 and CD8 T cells. The third, chronic stage of infection, begins approximately 25–28 dpi, and is marked by further recruitment of effector T cells and a plateau in the number of bacteria in the lungs.

Here, we developed and applied new approaches to specifically quantitate live Mtb in different lung myeloid subsets in the early chronic stage of infection (28 dpi). We found that CD11c[lo] monocyte-derived cells we term MNC1 (mononuclear cell subset 1) harbor 4 to 6-fold more live bacteria per infected cell than other lung cell subsets. We used RNA sequencing (RNA-seq) to identify differentially expressed genes and pathways that distinguish highly permissive MNC1 cells from other lung myeloid subsets. This revealed that MNC1 express lower levels of lysosome biogenesis genes compared to AM, a finding we confirmed at the protein and functional levels. Furthermore, activation of lysosome function by the c-Abl tyrosine kinase inhibitor nilotinib improved control of Mtb in vitro and in vivo. Our findings indicate that Mtb recruits and exploits lysosome-poor cells for persistence, and enhancing lysosome abundance and function is a potential strategy to combat Mtb infection.

## Results

### Characterization of lung cell populations containing Mtb after aerosol infection

We previously used flow cytometry to identify and characterize the lung leukocyte subsets containing fluorescent protein-expressing Mtb and found that at ≥ 14 dpi, the bacteria were found predominantly in neutrophils and in two subsets of monocyte-derived cells which we termed 'recruited macrophages' and 'myeloid dendritic cells' [12]. Since subsequent studies have identified markers that allow higher resolution definition of lung leukocytes [9], we repeated those studies, using flow cytometry to analyze lung cells from mice infected with Mtb H37Rv expressing ZsGreen at 14, 21, and 28 dpi (Figs 1A–1C and S1). The results were consistent with reports that the number of AM changed minimally over the 28 days of infection, while the number of neutrophils, MNC1, and MNC2 in the lungs markedly increased by 21–28 dpi (Fig 1A). As we previously reported [12,22], the number of infected (ZsGreen[+]) neutrophils and MNC2 exhibited a transient peak 21 dpi, followed by a decrease by 28 dpi. In contrast, the number of infected MNC1 cells increased from 14 dpi to 28 dpi (Fig 1B). When considered as the fraction of the total number of Mtb-infected cells, AM were dominant 14 dpi consistent with prior results [5,7,10]. By 21 dpi, AM constituted a minor fraction of the total,

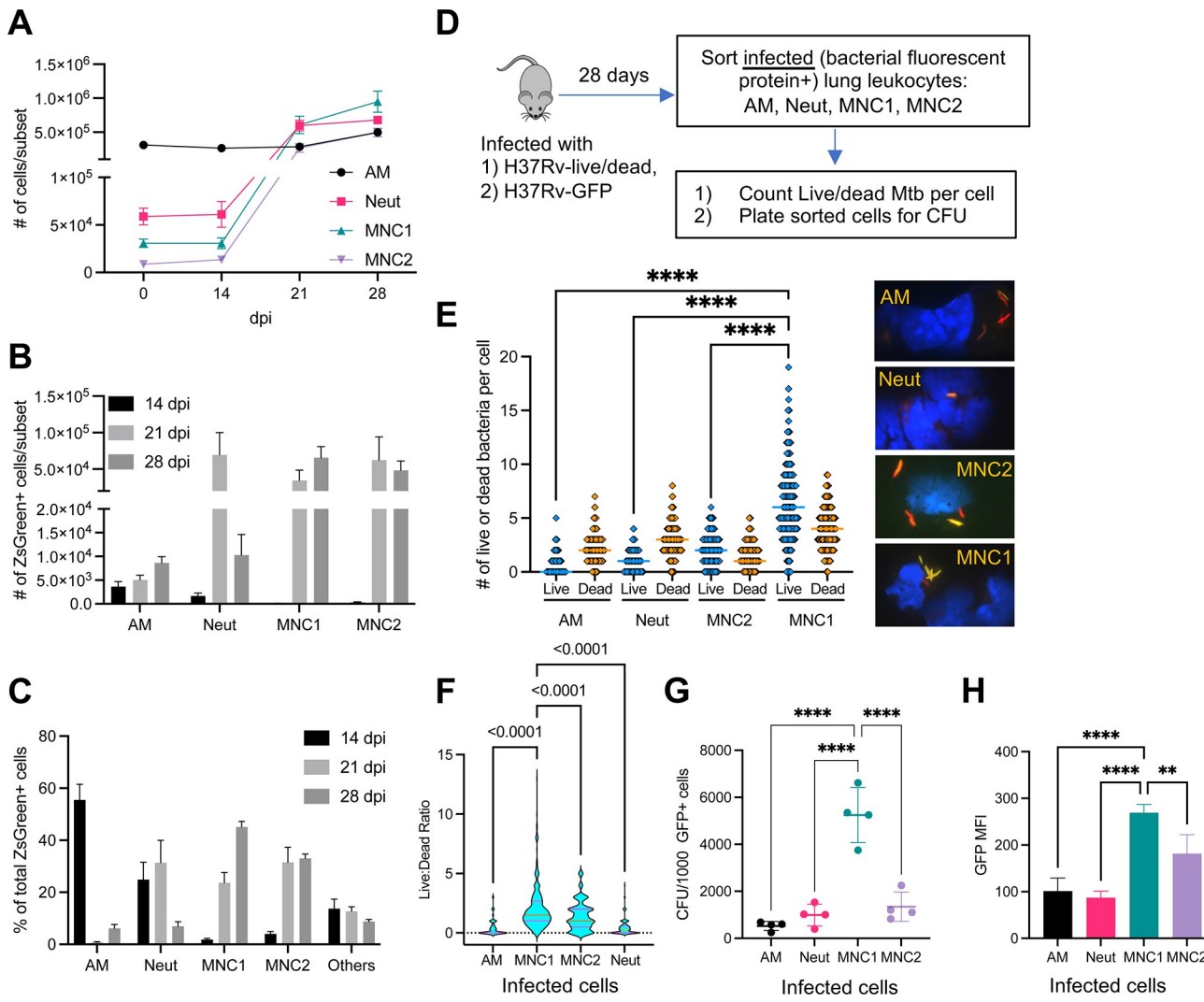

**Fig 1. MNC1 are highly permissive for Mtb intracellular survival.** C57BL/6 mice were infected with the designated strain of Mtb by low-dose aerosol. Lung cells were isolated at the indicated time points post infection. (A) Lung phagocyte population dynamics after Mtb (H37Rv-ZsGreen) infection. Neutrophils (Neut), MNC1, and MNC2 increase in response to Mtb infection, while the number of alveolar macrophages (AM) changed minimally (n = 4–5). (B) Number of Mtb H37Rv-ZsGreen+ cells in distinct lung phagocyte subsets by flow cytometry (n = 4–5). (C) Cell type composition of total Mtb H37Rv-ZsGreen+ lung leukocytes by flow cytometry. After early predominant distribution of Mtb in AM and neutrophils, MNC1 and MNC2 dominate by 28 dpi (n = 4–5). (D) Schematic diagram of procedures to quantitate intracellular live Mtb in sorted lung phagocyte subsets. C57BL/6 mice (n = 10) were infected with Mtb H37Rv-live/dead or H37Rv-GFP and cells containing fluorescent protein-expressing bacteria were analyzed at 28 dpi. (E) MNC1 contain the largest number of live Mtb per cell 28 dpi. Quantitation of live (GFP+mCherry+) or dead (GFP-mCherry+) Mtb per infected cell (n ≥ 300) was performed by fluorescence microscopy on viable cells sorted from mice infected with Mtb H37Rv-live/dead. Representative images on the right show live and dead Mtb. Dead (mCherry+GFP-) Mtb are red; live (mCherry+GFP+) appear yellow. The majority of the Mtb in AM and neutrophils are dead, while the majority of the bacteria in MNC1 and MNC2 are live (MNC1>MNC2). Images were taken using a 63x oil objective. (F) The ratio of live to dead bacteria in individual cells was calculated from the raw data used for Fig 1E. The orange horizontal bar indicates the median and the pink horizontal bars indicate the 25th and 75th percentiles. Statistical comparisons used the Kruskal-Wallis test with Dunn's multiple comparisons test. (G) MNC1 contain the largest number of live Mtb (H37Rv-GFP) at 28 dpi. Cells in each subset were sorted according to surface phenotypes and for bacterial status (GFP+). CFU of sorted GFP+ cells in each subset were counted after 3 wk of incubation. The results are expressed as CFU per 1000 GFP+ cells in each subset. (H) GFP MFI correlates with live Mtb burdens in the 4 infected lung myeloid cell subsets from mice infected with H37Rv-GFP (28 dpi). Results are presented as mean ± SD. Representative data from two independent experiments are shown for (A-C, G-H). **p<0.01 ****p<0.0001 by one-way ANOVA for (E, G, H).

and neutrophils, MNC1, and MNC2 were the predominant populations that contained Mtb (Fig 1C). At this time point, MNC2 were the dominant subset of infected cells (Fig 1C), consistent with a recent report that used sfYFP-expressing Mtb and a similar flow cytometry scheme [8]. However, MNC1 expanded further as a fraction of the infected cells at 28 dpi. It is also notable that regardless of the cell subset, only a minority of the cells in a subset contain Mtb. Overall, the data confirm that AM are the primary Mtb-infected population in the initial stage of aerosol infection, and that MNC1 and MNC2 are the major infected cell subsets after the initial stage.

MerTK and CD64 have been used as markers to define CD11b<sup>lo</sup> AM and CD11b<sup>+</sup> IM in various contexts, including mice intranasally infected with a high dose of Mtb [7,23–25]. Since strong evidence indicates that MNC1 and MNC2 are derived from monocytes [8,9,15], we used a modified flow panel (S2A Fig) to query if MNC1 and MNC2 subsets are similar to IM defined using MerTK and CD64 expression. This revealed that ~95% of AM were MerTK<sup>+</sup>CD64<sup>+</sup>, while only 10–22% of MNC1 and 20–47% of MNC2 were MerTK<sup>+</sup>CD64<sup>+</sup> from 14–28 dpi (S2B Fig). When we gated on Mtb<sup>+</sup> cells, we found that 70–97% of infected AM were MerTK<sup>+</sup>CD64<sup>+</sup>, while fewer MNC1 (10%-60%) and MNC2 (25%-86%) were MerTK<sup>+</sup>CD64<sup>+</sup>(S2C Fig). These results indicate that both MNC1 and MNC2 contain macrophage-like cells and that the use of MerTK and CD64 to define macrophages excludes a significant fraction of the cells that harbor Mtb in the lungs.

## Cell subset distribution of live Mtb during the chronic stage of infection

Use of fluorescent protein-expressing strains of Mtb coupled with flow cytometry has been invaluable in revealing the diversity of the cell types that are infected in vivo, but these procedures alone do not reveal the viability of the intracellular bacteria. Likewise, although there is evidence that cells exhibiting brighter bacterial fluorescence harbor more bacteria [8], a given cell can contain both live and dead Mtb that contribute to the fluorescence signal. To quantitate live intracellular Mtb, we utilized Mtb H37Rv carrying a live/dead reporter plasmid that drives constitutive expression of mCherry fluorescent protein (all bacteria) and doxycycline-inducible expression of green fluorescent protein (GFP; only live bacteria) [26]. Using fluorescent cell sorting and fluorescence microscopic evaluation, we enumerated live (mCherry<sup>+</sup>GFP<sup>+</sup>) and dead (mCherry<sup>+</sup>GFP<sup>-</sup>) bacteria in individual infected cells (Fig 1D).

We sorted four lung cell subsets from mice infected with live/dead-H37Rv (28 dpi) and examined mCherry<sup>+</sup> Mtb in flow-sorted AM, neutrophils, MNC1, and MNC2 by fluorescence microscopy. This revealed that the majority of the bacteria in mCherry<sup>+</sup> AM were dead (GFP<sup>-</sup>) at this time point (Fig 1E). Within the mCherry<sup>+</sup> AM population, there was considerable variation in the number of total bacteria (range: 1–7) per cell. While most of the mCherry<sup>+</sup> AM contained both live and dead bacteria, some contained only dead bacteria and others contained only live bacteria. Overall, dead bacteria were more abundant than live bacteria in AM. These findings are consistent with the results indicating that Mtb expansion in AM appears to be arrested by 21 dpi [8]. Similar results were apparent in the neutrophil population, in which there was also a range of bacteria per mCherry<sup>+</sup> cell; most contained both live and dead bacteria, with dead bacteria predominating.

In contrast to the bacterial states in AM and neutrophils, in both monocyte-derived cell subsets, MNC1 and MNC2, live bacteria were more abundant than dead bacteria (Fig 1E). Some MNC2 cells contained single live or dead bacteria, while most contained multiple bacteria, including both live and dead bacteria in the same cell. Since the MNC2 cell subset resembles the CD11c<sup>hi</sup> cell subset previously reported to represent the largest fraction of YFP-expressing Mtb at 21 dpi [8], these results confirm that this (or a related) subset contains predominantly live bacteria at a later time point (28 dpi). The distribution of Mtb in the MNC1

subset was similar to that in the MNC2 subset, although MNC1 cells contained more live bacteria per mCherry$^+$ cell. Very few infected MNC1 cells contained single bacteria, while the median number of either live or dead bacteria per mCherry$^+$ cell exceeded that observed in any of the other cell subsets, including MNC2. Notably, the median number of live Mtb per mCherry$^+$ MNC1 cell (6; range, 0–19) was higher than that of the dead bacteria (4; range, 0–9). Furthermore, the ratio of live to dead bacteria was higher in MNC1 cells compared to other subsets (Fig 1F). These results indicate that, although each of the sorted lung cell subsets exhibited the ability to kill some virulent Mtb, MNC1 cells are the least restrictive for intracellular growth of Mtb at a time point (28 dpi) when T cell responses are well developed and the total size of the bacterial population in the lungs has stabilized [18].

Since use of doxycycline could exert other activities on the bacteria and/or host cells, we performed a similar experiment using Mtb H37Rv constitutively expressing enhanced GFP [12], and quantitated live Mtb as colony-forming units (CFU) present in sorted GFP$^+$ cells in each of the subsets (Fig 1D). This revealed that 28 dpi, AM and neutrophils both contained ~400–600 CFU/1,000 GFP$^+$ cells, MNC2 contained approximately 1,000 CFU/1,000 GFP$^+$ cells, and MNC1 contained 4,000–6,000 CFU/1,000 GFP$^+$ cells (Fig 1G). The GFP or ZsGreen median fluorescence intensity (MFI) of infected MNC1 was higher than the other infected subsets (Figs 1H and S3A), correlating with CFU and the number of live Mtb per cell in each infected subset. In ex vivo studies, we determined that MNC1 and MNC2 have a similar Mtb phagocytosis capacity, but lower than AM and neutrophils, suggesting that the high Mtb burden in MNC1 is not due to a higher phagocytosis activity (S3B Fig). Moreover, at 56 dpi, Mtb resided predominantly in MNC1, which also harbored more Mtb per cell than other subsets as indicated by bacterial fluorescence (S3C and S3D Fig).

Together, these results suggest that during the chronic stage of infection, after the development of adaptive T cell responses, AM and neutrophils restrict and kill virulent Mtb effectively, although they do harbor some live bacteria. In contrast, MNC1 and MNC2 are permissive for Mtb, as they predominantly harbor live bacteria.

## RNA-seq analysis of lung myeloid subsets reveals diversity and differential gene and pathway expression

To identify mechanisms that potentially account for the differential ability of lung cell subsets to restrict and kill Mtb during chronic infection, we carried out RNA-seq analysis for 8 cell populations sorted from the lungs of Mtb-infected mice: AM, neutrophils, MNC1, and MNC2; each either infected or bystander (uninfected). Although we found differentially expressed genes between infected and bystander cells for each subset (S4 Fig), a t-stochastic neighbor embedded (tSNE) plot of the RNA-seq data showed segregation of the four cell subsets (Fig 2A). In this analysis, Mtb-infected cells were subclusters within each cell subset, indicating that the presence of intracellular bacteria does not determine the cell phenotype. The transcriptome data revealed that canonical markers *Ear1*, *Mrc1*, *Pparg*, *Siglecf*, *Siglec1*, and *Fabp4* were indeed highly expressed by AM, and neutrophils expressed *Il1r2*, *Csf3r*, *Cxcr2*, *S100a8*, *S100a9* and *Retnlg* (Fig 2B). In contrast, monocyte markers such as *Ccr2*, *Cx3cr1*, *Mafb*, *Ly6c1* and *Tfrc* were expressed at a significantly higher level in MNC1 and MNC2 compared to AM, consistent with other evidence that MNC cells are derived from monocytes [9]. MNC2 also expressed transcripts characteristic of DC including *Ccl22*, *Il12b*, *Flt3*, *Ccr7*, *Cd83*, and *Cd86*. However, other studies have indicated that classical DC are much less abundant than monocyte-derived cells in the lungs of Mtb-infected mice [23], and that CD11c$^{hi}$ monocyte-derived cells (MNC2 in this study) also express transcripts characteristic of macrophages [8], indicating that this population may be heterogenous; we are continuing to seek markers that improve the discrimination of these cell subsets.

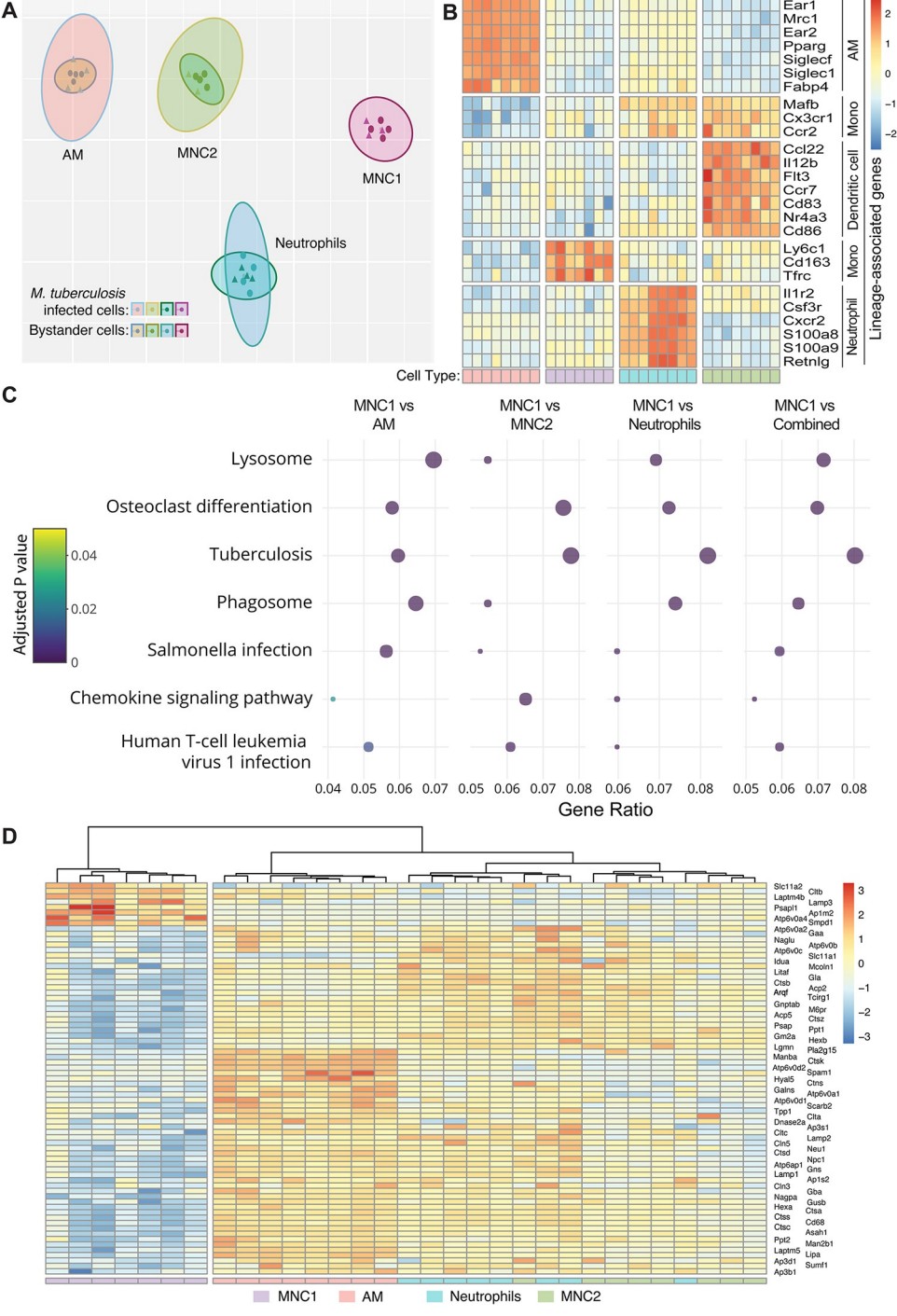

**Fig 2. RNA-seq analysis of live-sorted phagocyte subsets from lungs of Mtb-infected mice reveals evidence of deficient lysosome biogenesis in MNC1.** C57BL/6 mice (n = 20) were infected with Mtb H37Rv-mCherry by low dose aerosol. After 28 days, 10,000 live cells from each subset in each of the 4 pools (5 infected mice per pool) were sorted directly into RNAlater and processed for bulk RNA-seq. (A) t-stochastic neighbor embedding (t-sne) plot showing distinct clusters of four myeloid cell types based on RNA sequencing on cells sorted from lungs of Mtb H37Rv-mCherry infected mice. Within a clustered subset there is substantial overlap between Mtb infected and bystander cells after exclusion of one infected MNC1 outlier sample. (B) Heatmap showing separation of four distinct cell types based on Z-scores from variance stabilized read counts for lineage markers. The color coding of the cell types shown at the bottom corresponds to the colors in (A). (C) Dot plot showing 7 of 18 KEGG pathways that differ significantly with an enrichment ratio greater than 0.04 for AM, MNC2, neutrophils, combined analysis (AM, MNC2, neutrophils) and

MNC1. The color represents the adjusted p values, the graph is ordered by descending values (lowest p value = 1.5 x $10^{-16}$ for the lysosome pathway, combined analysis), while the dot size is proportional to the gene count. (D) Heatmap of the Z-scores from variance stabilized read counts for significantly differentially expressed genes of KEGG lysosome pathway shows separation of MNC1 from AM, MNC2, and neutrophils.

We performed KEGG pathway analysis to identify differences contributing to the differential Mtb permissiveness of MNC1 compared with the other 3 lung cell subsets. We identified the pathways that exhibited >2-fold enrichment with a Benjamini-Hochberg adjusted p<0.05 (Figs 2C and S5). Of these, the "Lysosome" pathway was differentially expressed in MNC1 compared with the other subsets (Fig 2C). Genes of lysosome pathways encode lysosomal membrane proteins (e.g., LAMP1, LAMP2), lysosomal hydrolases (e.g., cathepsin proteases and glycosidases), and lysosome vacuolar proton ATPase (V-ATPase) subunits (Fig 2D). These genes were not differentially expressed in Mtb-infected cells compared to bystander cells within each subset, and 65 of 73 genes were underexpressed in both Mtb-infected and bystander MNC1 cells. Among the underexpressed lysosome genes, beta-hexosaminidase (HEXB), cathepsins B, S, and L, and phospholipase A2 (PLA2G15) have been reported to contribute to antimycobacterial activity [27–29]. Notably, some of the underexpressed lysosome genes such as V-ATPase subunits are also important components of the KEGG "Tuberculosis" and "Phagosome" pathways. Thus, we hypothesized that intrinsic deficiency of lysosome biogenesis in MNC1 is a mechanism that contributes to their Mtb permissiveness.

## Mtb-permissive MNC1 are deficient in lysosomal enzyme activity

Since Mtb-restrictive AM and Mtb-permissive MNC1 cells exhibit the greatest difference in their expression of genes involved in lysosome biogenesis, and since Mtb survives in macrophages at least in part by limiting lysosome-dependent phagosome maturation (reviewed in [2]) and lysosome-dependent autophagy [30–33], we quantitated lysosome activities and content in AM, MNC1 and MNC2.

We first used a fluorogenic Cathepsin B (CTSB) assay and flow cytometry to compare the enzymatic activity of CTSB in AM, MNC1 and MNC2 from lungs of Mtb-infected mice (28 dpi). In this assay, the substrate fluoresces after cleavage and can be quantitated by fluorescence microscopy or flow cytometry as a reflection of cathepsin B enzymatic activity [34]. When incubated with the CTSB substrate, AM showed ~7-fold higher MFI of the product than did MNC1 (Fig 3A and 3B). We then sorted cells from Mtb-infected mice, incubated them with the CTSB substrate, and quantitated fluorescence by microscopy. In this assay, AM incubated with the substrate exhibited red fluorescence that was readily detectable by fluorescence microscopy (Fig 3C, left panels). In contrast, sorted MNC1 cells exhibited barely detectable fluorescence. As a control, the specific V-ATPase inhibitor bafilomycin A1 that prevents lysosomal acidification and lysosomal cathepsin activity [35], blocked generation of fluorescence in AM (Fig 3C, right panels). Quantitative image analysis confirmed ~3-4-fold higher CTSB product fluorescence in AM compared with MNC1 (Fig 3D).

Since RNA-seq analysis revealed decreased MNC1 expression of mRNA encoding other lysosomal cathepsins (H, Z, K, D, A, S, L, and C) (Fig 2D and S1 Data), we analyzed additional lysosomal cathepsin activities using a pool of substrates for cathepsins B, K, and L. This yielded results similar to those obtained with the CTSB substrate: fluorescent product generation was ~3-fold greater in AM than in MNC1 cells, and the fluorescence generation in AM was abrogated by bafilomycin A1 (Fig 3E and 3F). Notably, MNC2 showed an intermediate lysosome activity in all assays shown in Fig 3. Together, these results provide functional evidence for the lower expression of lysosomal cathepsin mRNAs in MNC1 compared with AM.

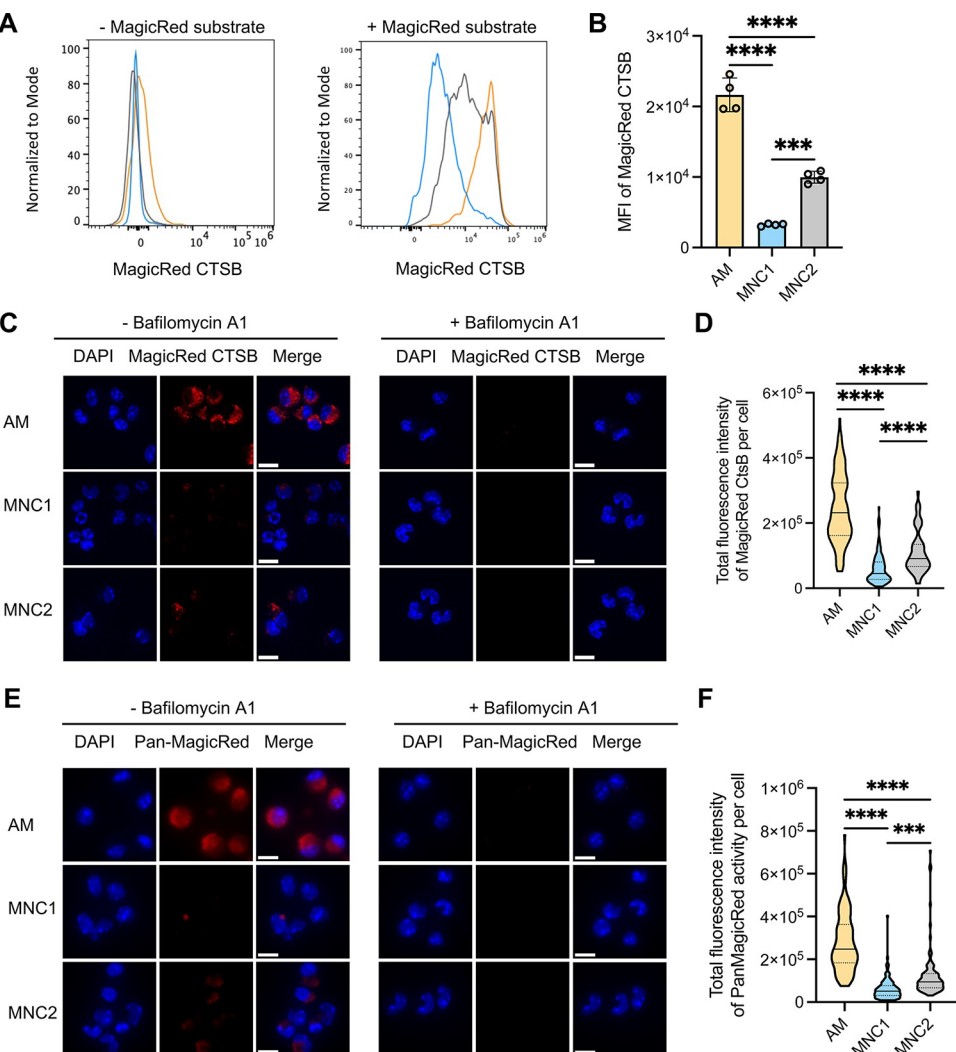

**Fig 3. MNC1 cells are deficient in lysosomal cathepsin proteolytic activities.** C57BL/6 mice were infected by aerosol with ~100 Mtb H37Rv-ZsGreen or H37Rv-mCherry. At 28 dpi, mouse lungs were harvested for analysis. (A-B) Representative histograms and MFI of the fluorescent product of CTSB activity for AM and MNC1. Lung cells from mice infected with H37Rv-ZsGreen (28 dpi, n = 4) were incubated with MagicRed CTSB substrate for 30 min, and stained with antibodies for discrimination of cell subsets, followed by flow cytometry analysis. (C) CTSB activities in cells sorted from mice (n = 10) infected with H37Rv-mCherry for 28 days, analyzed by confocal microscopy. Live-sorted cells from each cell subset were treated with the fluorogenic CTSB substrate in the absence or presence of Bafilomycin A1 (BafA, 100 nM) for 1 h. Cells were fixed and analyzed by confocal microscopy. Images were taken using a 100x oil objective. Scale bars, 10 μm. (D) Quantification of fluorogenic CTSB product fluorescence per cell from the left panel in (C). >100 cells per subset were analyzed using ImageJ. (E) Pan-cathepsin activities in cells sorted from lungs of mice (n = 10) infected with H37Rv-mCherry (28 dpi). Sorted cells were treated with a pool of MagicRed fluorogenic cathepsin substrates (CTSB+CTSK+CTSL) in the absence or presence of 100 nM BafA for 1 h. Images were taken using a fluorescence microscope with a 100x oil objective. Scale bars, 10 μm. (F) Quantification of Pan-cathepsin product fluorescence per cell from the left panel of (E). >100 cells per subset were analyzed using ImageJ. ***p<0.001, ****p<0.0001 by one-way ANOVA (B, D, F). Results are presented as mean ± SD. Representative data from two independent experiments are shown for (A-D).

## Mtb-permissive MNC1 are deficient in lysosomal acidification

Lysosomal hydrolases and antimicrobial activities [36,37] require an acidic environment for their functions; the acidic lysosome environment is provided by the activity of a V-ATPase that comprises multiple V0 and V1 protein subunits [38]. Although the mRNA level of

V-ATPase subunits did not differ significantly between infected and bystander cells, RNA-seq revealed lower expression of multiple V-ATPase subunits in MNC1 compared with AM and MNC2 except for Atp6v0e2, Atp6v0a4, Atp6v1b1 and Atp6v1c2, which may be regulated differently (Fig 4A). Since vacuolar acidification requires assembly of multimers that incorporate all of the V-ATPase subunits, we hypothesized that MNC1 are deficient in lysosomal acidification, contributing to their deficient lysosome activity.

We first analyzed the protein level of V-ATPase subunits in AM and MNC1 isolated from the lungs of Mtb-infected mice (28 dpi). By flow cytometry, intracellular ATP6V1B2 was ~3 fold higher in AM than in MNC1 (Fig 4B). As assessed by immunofluorescence microscopy on sorted AM and MNC1, all of the V-ATPase subunits that we examined (ATP6V1B2 and ATP6V0D2) were present at 2–4 fold higher levels in AM compared with MNC1 (Fig 4C). These results are consistent with the results of RNA analyses, indicating that MNC1 may be less capable of lysosome and phagolysosome acidification compared with AM.

To determine whether the lower abundance of V-ATPase subunits has functional significance for lysosome acidification, we quantitated accumulation of the anionic (pKa = 9.84) fluorochrome, cresyl violet, which labels acidic compartments [39]. Fluorescence microscopy revealed marked accumulation of cresyl violet in punctate structures in AM, but minimal accumulation in MNC1 cells (Fig 4D and 4E). Treatment of lung cells with bafilomycin A1 to block lysosome acidification abrogated accumulation of cresyl violet (Fig 4D and 4F), confirming that cresyl violet accumulation and fluorescence are the consequence of lysosome acidification mediated by V-ATPase activity. As expected, MNC2 showed an intermediate level of V-ATPase expression and lysosomal acidification. These results suggest that low expression of V-ATPase subunits contributes to poor lysosomal acidification in MNC1.

## Mtb-permissive MNC1 are deficient in lysosome enzyme content and LAMP1$^+$ organelles

The reduced lysosomal cathepsin activity in MNC1 compared with AM could be secondary to reduced lysosome acidification, or due to decreased abundance of lysosomal enzyme protein, or both. By immunofluorescence microscopy, we found abundant CTSB staining in a punctate distribution in AM, while CTSB-containing puncta were less numerous in MNC1, and this was substantiated by quantitative image analysis (Fig 5A and 5B). These findings indicate that deficient lysosome acidification alone is unlikely to account for the lower cathepsin activity in MNC1 compared with AM, and, consistent with lower mRNA expression, immunoreactive CTSB protein is also less abundant in MNC1 than in AM.

We then quantitated the abundance of the lysosomal (and late endosomal) membrane protein, LAMP1 by immunofluorescence microscopy on cells sorted from lungs of infected mice. In line with the RNA-seq data, this revealed abundant LAMP1 punctate fluorescence throughout the cytoplasm of AM (Fig 4I and 4J). In contrast, LAMP1 staining of MNC1 cells was less intense, although the distribution, size, and shape of the LAMP1$^+$ puncta resembled those in AM (Fig 5C and 5D). We then quantitated intracellular LAMP1 by flow cytometry and found that the MFI of intracellular LAMP1 was approximately 6-fold higher in AM than in MNC1 cells, consistent with fewer LAMP1$^+$ lysosomes or lower LAMP1 content per organelle in MNC1 (Fig 5E). Again, MNC2 showed an intermediate level of CSTB protein and LAMP1$^+$ organelles (Fig 5A–5E). These results indicate that MNC1 are deficient in classically defined lysosomes.

Using a fluorescent dextran pulse-chase assay, we found that compared with MNC1, the MFI of dextran fluorescence in AM was 12-fold higher, while the MFI of dextran fluorescence in MNC2 cells was intermediate between that of alveolar macrophages and that of MNC1 cells

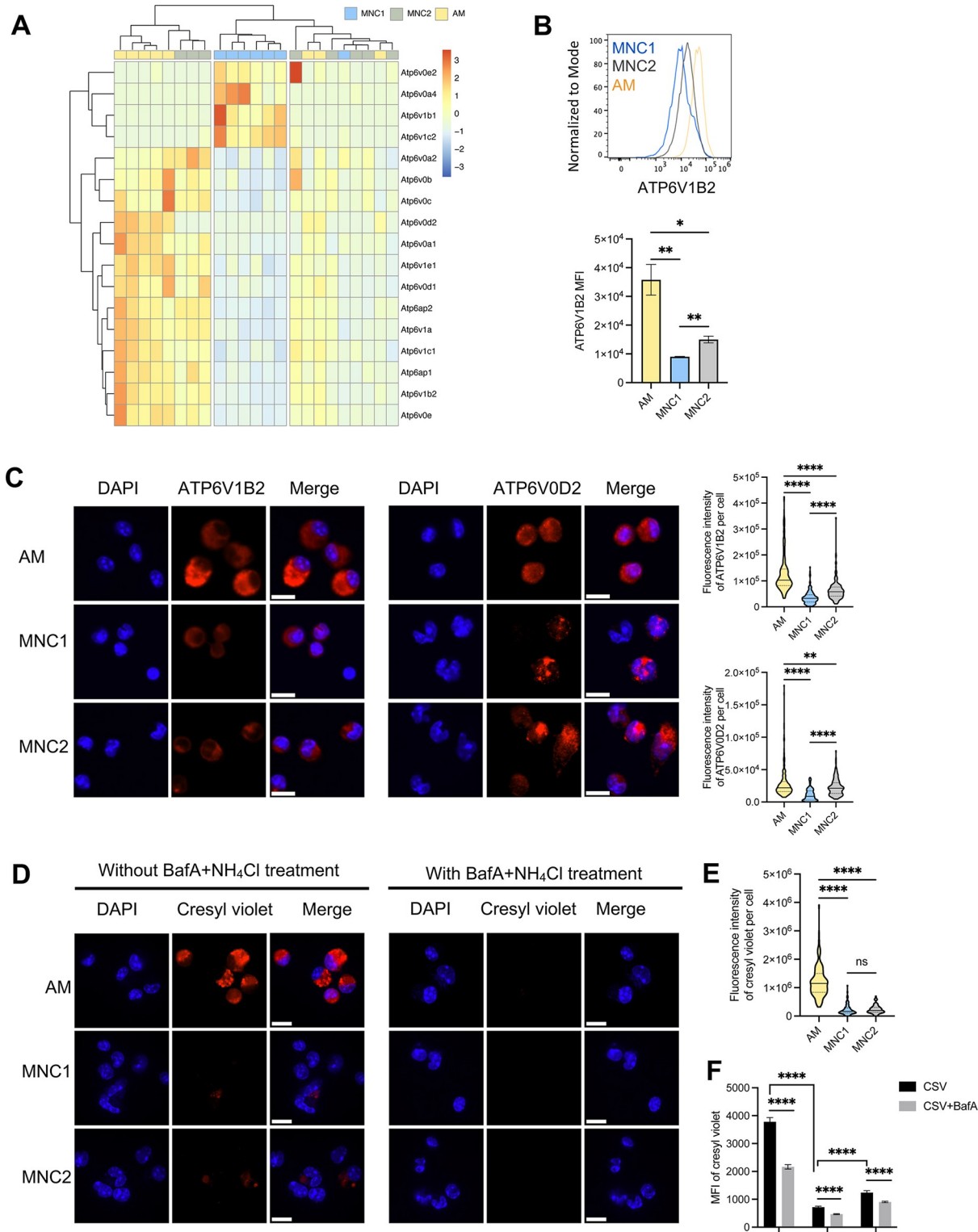

**Fig 4. MNC1 cells exhibit defective lysosome acidification.** C57BL/6 mice were infected by aerosol with ~100 Mtb H37Rv-ZsGreen or H37Rv-mCherry. At 28 dpi, mouse lungs were harvested for analysis. (A) Heatmap of the Z-scores from variance stabilized read counts for differentially expressed V-ATPase subunit genes in AM, MNC1 and MNC2. (B) Representative histograms and MFI of ATP6V1B2 protein immunostaining by flow cytometry of fixed and permeabilized cell subsets from mice infected with H37Rv-ZsGreen (28 dpi, n = 5). (C) Immunofluorescence analysis of four V-ATPase subunits in cells sorted from H37Rv-mCherry-infected mice (28 dpi, n = 10). Representative

images were taken by a fluorescence microscope with a 40x oil objective. Scale bars, 10 μm. Quantification of V-ATPase subunit fluorescence per cell from >100 cells per subset was done using ImageJ. (D) Cresyl violet assay of lysosome acidification in cells sorted from lungs of mice infected with H37Rv-mCherry (28 dpi, n = 10). Sorted cells were treated with 5 μM cresyl violet in the absence or presence of BafA (100 nM) and NH$_4$Cl (10 mM) for 30 min. Images were taken using a confocal microscope with a 100x oil objective. Scale bars, 10 μm. (E) Quantification of cresyl violet fluorescence per cell for the left panel in (D) using ImageJ (>100 cells of each type). (F) Representative histograms and MFI of the cresyl violet (CSV) signal for AM and MNC1 (28 dpi, n = 4). Lung single cells were incubated with 2 μM cresyl violet in the absence or presence of BafA (200 nM) for 30 min in the cell incubator. (G) Immunofluorescence analysis of CTSB protein in cells sorted from H37Rv-mCherry-infected mice (28 dpi, n = 10). Representative images were taken using a confocal microscope with a 100x oil objective. Scale bars, 10 μm. (H) Quantification of CTSB protein immunostaining MFI per cell for (G) by ImageJ (>100 cells per subset). (I) Immunofluorescence analysis of LAMP1 in cells sorted from H37Rv-mCherry-infected mice (28 dpi, n = 10). Representative images were taken by a confocal microscope with a 100x oil objective. Scale bars, 10 μm. (J) Quantification of LAMP1 MFI per cell for (I) by ImageJ (>100 cells per subset). (K) Representative histograms and MFI of intracellular LAMP1 analyzed by flow cytometry for AM and MNC1 from mice infected with Mtb H37Rv-ZsGreen (28 dpi, n = 5). *p<0.05, **p<0.01, ****p<0.0001 by one-way ANOVA (B-C, E) or unpaired Student's t test (F). Data are presented as mean ± SD. Representative data from 2–3 independent experiments are shown for (B, D- E).

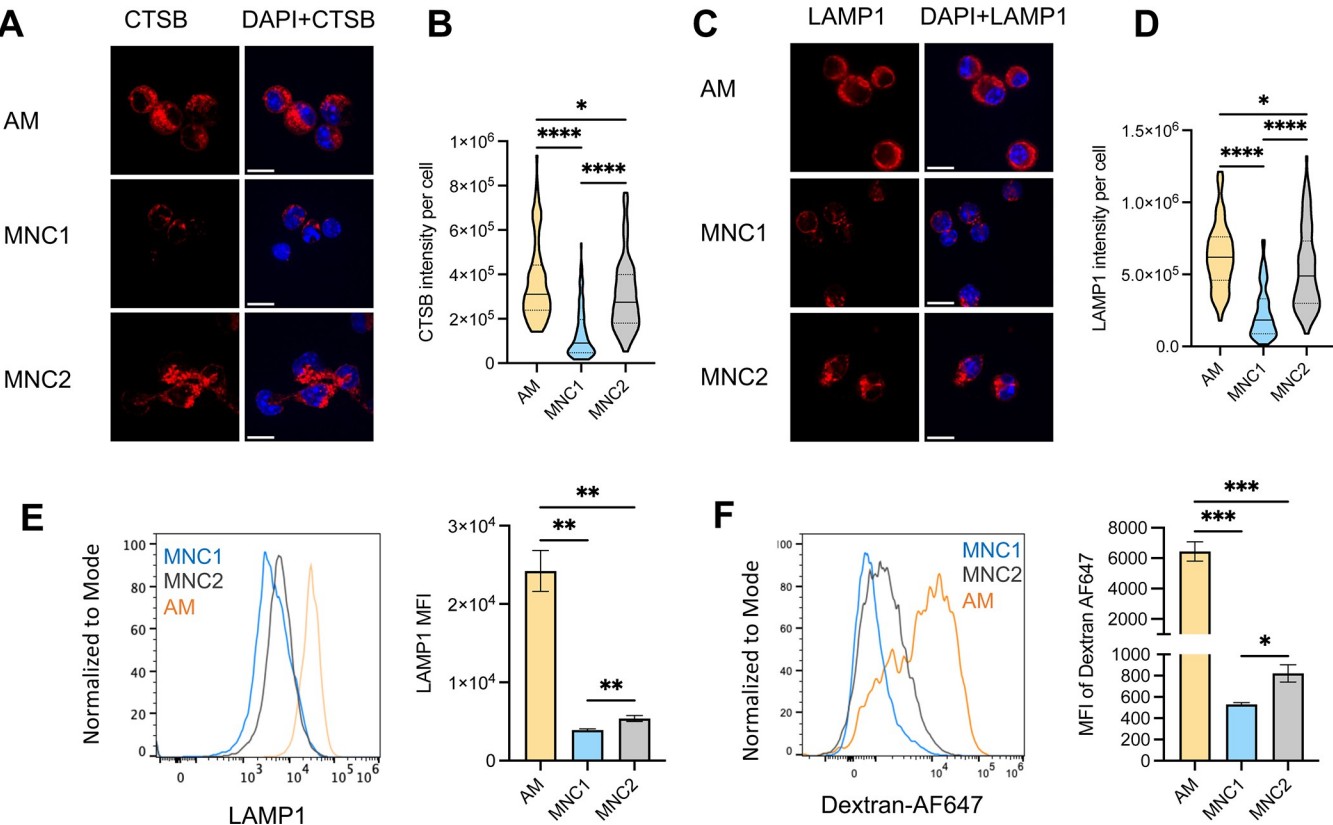

**Fig 5. MNC1 are deficient in lysosomal cathepsin B and LAMP1.** C57BL/6 mice were infected by aerosol with ~100 Mtb H37Rv-ZsGreen or H37Rv-mCherry. At 28 dpi, mouse lungs were harvested for analysis. (A) Immunofluorescence analysis of CTSB protein in cells sorted from H37Rv-mCherry-infected mice (28 dpi, n = 10). Representative images were taken using a confocal microscope with a 100x oil objective. Scale bars, 10 μm. (B) Quantification of CTSB protein immunostaining fluorescence per cell for (G) by ImageJ (>100 cells per subset). (C) Immunofluorescence analysis of LAMP1 in cells sorted from H37Rv-mCherry-infected mice (28 dpi, n = 10). Representative images were taken by a confocal microscope with a 100x oil objective. Scale bars, 10 μm. (D) Quantification of LAMP1 fluorescence per cell for (I) by ImageJ (>100 cells per subset). (E) Representative histograms and fluorescence of intracellular LAMP1 analyzed by flow cytometry for AM and MNC1 from mice infected with Mtb H37Rv-ZsGreen (28 dpi, n = 5). (F) Lung cells were pulsed with 20 μg/mL Dextran-Alexa Fluor 647 for 1h and chased for another 1h. Samples were then processed for flow cytometry analysis. *p<0.05, **p<0.01, ***p<0.001, ****p<0.0001 by one-way ANOVA (B, D, E, F). Data are presented as mean ± SD. Representative data from 2–3 independent experiments are shown for (A-E).

(Fig 5F). Although differences in the rate of endocytosis could yield similar results, these are concordant with those of our other assays of lysosome quantity and content and the combined data strongly support the conclusion that MNC1 cells are deficient in lysosomes when compared with the other cell subsets in the lungs of Mtb-infected mice.

In addition, MNC1 retained this phenotype at 56 dpi, as seen by lower intensity staining of both LAMP1 and ATP6V1B2 compared to AM, and MNC2 had an intermediate level of these proteins (S6 Fig). Together, these findings suggest that monocyte-derived cells, especially MNC1, in Mtb-infected lungs are defective in lysosome functions as indicated by deficiencies of lysosome abundance and lysosomal protein content compared with Mtb-restrictive AM.

## Differential expression and activation of the lysosomal regulator, TFEB, in AM and MNC1

Lysosome biogenesis and expression of genes whose products are involved in lysosome structure, acidification, and functions, are regulated by the transcription factor EB (TFEB) [40,41] through recognition of CLEAR (Coordinated Lysosomal Expression and Regulation) elements [42].

RNA-seq of live sorted lung cell populations revealed approximately 2-fold higher TFEB expression in AM compared to MNC1, while the TFEB mRNA level was not significantly different between infected and bystander cells (Fig 6A). TFEB localization and activity is regulated by phosphorylation, wherein phosphorylated TFEB is retained in the cytoplasm by binding to the cytoplasmic chaperone, 14-3-3, and dephosphorylated TFEB translocates to the nucleus, where it activates transcription of lysosome biogenesis genes [43]. Immunofluorescence staining and microscopy on sorted AM and MNC1 revealed heterogeneity in the distribution of TFEB between cytoplasm and nucleus in both cell types (Fig 6B). Despite the heterogeneity, TFEB MFI was ~ 2 fold higher in AM compared with MNC1 (Fig 6C). Since TFEB executes its function in the nucleus [41,43], we also quantified nuclear TFEB. This revealed that TFEB localized to the nucleus in most, if not all, AM, while TFEB was nearly exclusively localized to the cytoplasm in MNC1 cells (Fig 6C and 6D). Compared with AM and MNC1, MNC2 showed an intermediate level of TFEB mRNA and protein (Fig 6A–6D). These results demonstrate a lower level of TFEB activity in MNC1 cells compared with AM, which is reflected downstream by the expression of TFEB-regulated lysosome genes.

## Mtb ESX-1 is required for MNC1 recruitment but does not determine MNC1 lysosome deficiency

Mtb resides in phagosomes that do not mature efficiently to phagolysosomes, and this property is dependent on the Mtb ESX-1 Type VII secretion system [2]. Therefore, we considered the possibility that Mtb ESX-1 alters lysosome biogenesis in lung monocyte-derived cells to facilitate its persistence. To test this, we infected mice with each of three ZsGreen expressing strains: Mtb H37Rv, Mtb H37Rv:ΔRD1, and the vaccine strain *M. bovis* BCG. The latter two lack the RD1 locus encoding a key part of ESX-1. We first compared the protein levels of LAMP1 and ATP6V1B2 for subsets by measuring MFI using analytical flow cytometry. This revealed that there was no difference of LAMP1 or ATP6V1B2 MFI in MNC1 and neutrophils from naïve mice or mice infected with Mtb H37Rv, Mtb H37Rv:ΔRD1, or BCG (S7A and S7B Fig). Interestingly, the LAMP1 and ATP6V1B2 levels of AM and MNC2 were significantly higher in Mtb H37Rv-infected mice compared with Mtb H37Rv:ΔRD1-infected mice, BCG-infected mice, or naïve mice. Despite the infection conditions, LAMP1 and ATP6V1B2 MFI of AM remained the highest among the lung phagocyte subsets. Consistent with the above results, MNC1 had lower protein levels of LAMP1 and ATP6V1B2 compared with AM, regardless of mouse infection or bacterial strain status. We further used the fluorogenic CTSB assay to assess the lysosome activities of lung

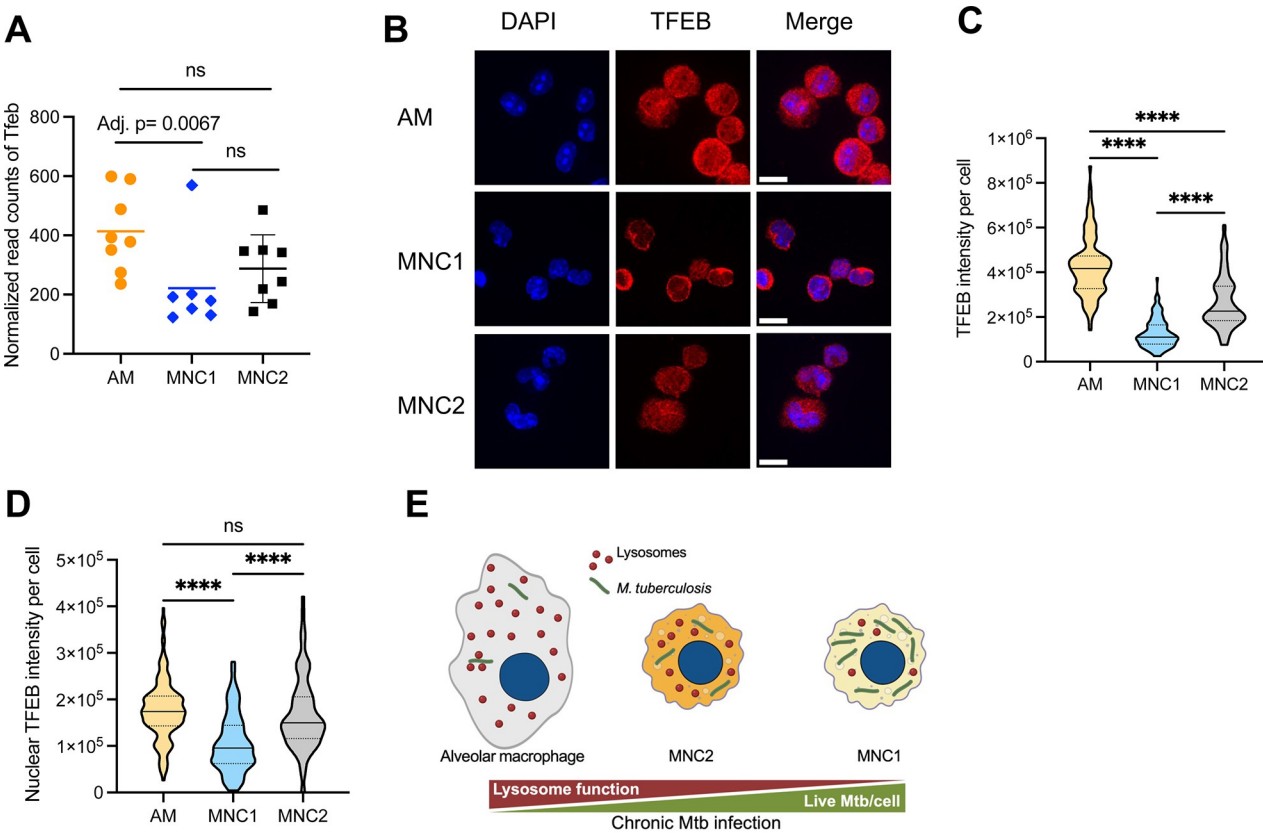

**Fig 6. TFEB in MNC1 is predominantly cytosolic.** C57BL/6 mice were infected with by aerosol with ~100 Mtb H37Rv-mCherry. At 28 dpi, lung cells were harvested for analysis. (A) RNA-seq data show that MNC1 express lower levels of *Tfeb* mRNA than do AM (28 dpi). The adjusted p-value shown is from the analyses of the dataset using DESeq2 as described in the methods section. DESeq2 employs the Wald test (n = 8 for AM and MNC2, n = 7 for MNC1). (B) AM have more nuclear TFEB than MNC1. Cells were isolated and sorted from lungs of mice infected with H37Rv-mCherry (28 dpi, n = 10). Anti-TFEB antibody was used for detecting TFEB in sorted cells of each subset. Representative images were taken by a confocal microscope with a 100x oil objective. Scale bars, 10 μm. (C-D) Quantification of total TFEB fluorescence and nuclear TFEB fluorescence per cell in (B) from >100 cells of each subset using ImageJ. (E) A model showing that Mtb restriction or survival depending on the functional lysosome abundance in distinct lung mononuclear cell subsets during chronic infection. ****p<0.0001 by one-way ANOVA (C, D). Data: mean ± SD. Representative data from two independent experiments are shown for (B-D).

cell subsets from mycobacteria-infected mice and naïve mice, where we observed results that were similar to analytical flow data for LAMP1 and ATP6V1B2 (S7C Fig). These data suggest that the defective lysosome functions of MNC1 are not determined by Mtb ESX-1.

In contrast, we did find that Mtb ESX-1 promotes the recruitment of MNC1, MNC2, and neutrophils to the lungs, and Mtb spread from AM to these cell subsets (S7D–S7F Fig), consistent with other results [13,44]. In line with these, mice infected with Mtb H37Rv:ΔRD1 or BCG showed lower lung bacterial burdens compared to the mice infected with Mtb H37Rv (S7G Fig). Together, these findings indicate that lysosome deficiency in MNC1 is not driven by Mtb ESX-1, although recruitment of lysosome-deficient permissive MNC1 is promoted by Mtb ESX-1. However, the differences in lung bacterial burdens in mice infected with wild type Mtb and ESX-1-deficient Mtb could be the primary determinant of cell recruitment.

### Pharmacological activation of TFEB and lysosomal function enhances control of Mtb

Since our findings suggest that Mtb exploits lysosome -deficient cells for persistence (Fig 6E), we investigated whether pharmacological activation of TFEB could increase lysosome

biogenesis and restriction of Mtb replication in macrophages. We tested multiple small molecules as potential activators of TFEB in murine bone marrow-derived macrophages (BMDM) by detecting nuclear-localized TFEB (S8A–S8C Fig). As expected, torin1, a potent inhibitor of mammalian target of rapamycin (mTOR) that regulates TFEB [45], induced TFEB nuclear translocation in BMDM (Fig 7A). Consistent with reports that the c-Abl tyrosine kinase inhibitor imatinib induces lysosomal acidification to inhibit Mtb growth in human monocyte-derived macrophages [46] and in Mtb-infected mice [47,48], we found that imatinib and an alternative c-Abl kinase inhibitor nilotinib activated TFEB nuclear translocation (Fig 7A) to an extent comparable to that of torin1 (Fig 7B). These results imply that c-Abl inhibitors activate TFEB and that this action accounts for certain of the earlier reports of the activity of imatinib.

We then determined if c-Abl inhibitors induce expression of TFEB downstream genes using quantitative PCR (qPCR). This revealed that imatinib and nilotinib significantly increased expression of *Tfeb*, *Lamp1*, *Ctsb*, *Ctsd*, *Hexa*, *Atp6v0d2* and *Atp6v1b2* (Fig 7C). Using siRNA knockdown, we found that induction of lysosome genes by nilotinib mainly depends on TFEB, and not TFE3, another regulator of lysosome biogenesis in macrophages [49] (Fig 7D). In line with the mRNA findings, treatment of BMDM with these TFEB activators increased LAMP1 and ATP6V1B2 at the protein level (Fig 7E). We then tested if these TFEB activators induce lysosomal acidification in BMDM. We observed that imatinib or nilotinib significantly increased cresyl violet fluorescence after 24-h treatment, and further increased after 72-h treatment (Fig 7F). In addition, CTSB enzyme activity increased in Mtb-infected BMDM treated with imatinib or nilotinib (Fig 7G). Interestingly, the increase of CTSB activity by nilotinib depended on both TFEB and TFE3 (Fig 7H); in agreement with this, imatinib and nilotinib also induced TFE3 nuclear translocation (S9 Fig), suggesting that TFEB and TFE3 may regulate distinct lysosome genes differently. In line with previous results in human primary macrophages [46], imatinib increased colocalization of Mtb with lysosomes (Fig 7I and 7J), and a similar result was obtained for nilotinib. Consistent with results of others [46,50], we verified that imatinib reduced bacterial loads in Mtb-infected BMDM (Fig 7K). We obtained similar results with nilotinib, a more specific and potent c-Abl inhibitor, indicating that the increase of lysosome activity and acidification promotes antimycobacterial activity of cultured macrophages. We found that the optimal effect of nilotinib on restricting intracellular Mtb requires TFEB and TFE3 (Fig 7L). The antimycobacterial activity of these agents was not due to direct action on Mtb, as growth in 7H9 medium was unaffected (S8D Fig), however both inhibitors interfered with a luciferase-based assay of mycobacterial viability. Collectively, these results suggest that activation of TFEB-mediated lysosomal acidification and activity enhances control of intracellular Mtb in macrophages.

## Nilotinib activates lysosomal functions of permissive myeloid subsets and reduces lung bacterial burden

Lysosomal activities are required for control of intracellular mycobacteria in cultured macrophages [27–29], and the above results and similar findings of others suggest that activation of lysosome biogenesis improves control of Mtb in macrophages [45,51–56]. Therefore, we asked whether c-Abl inhibitors can enhance Mtb restriction and the lysosomal functions of monocyte-derived cells in vivo (Fig 8A). Nilotinib was chosen for the in vivo study as it is more potent than imatinib in activating lysosome function and restricting intracellular Mtb replication (Fig 7E–7G and 7K). We found that nilotinib administration significantly reduced lung bacterial burden without affecting mouse body weight (Figs 8B and S10A), in line with the previous report that imatinib treatment enhanced control of Mtb in mouse lungs [47,48]. Although nilotinib treatment reduced the number of infected AM, MNC1, MNC2, neutrophils

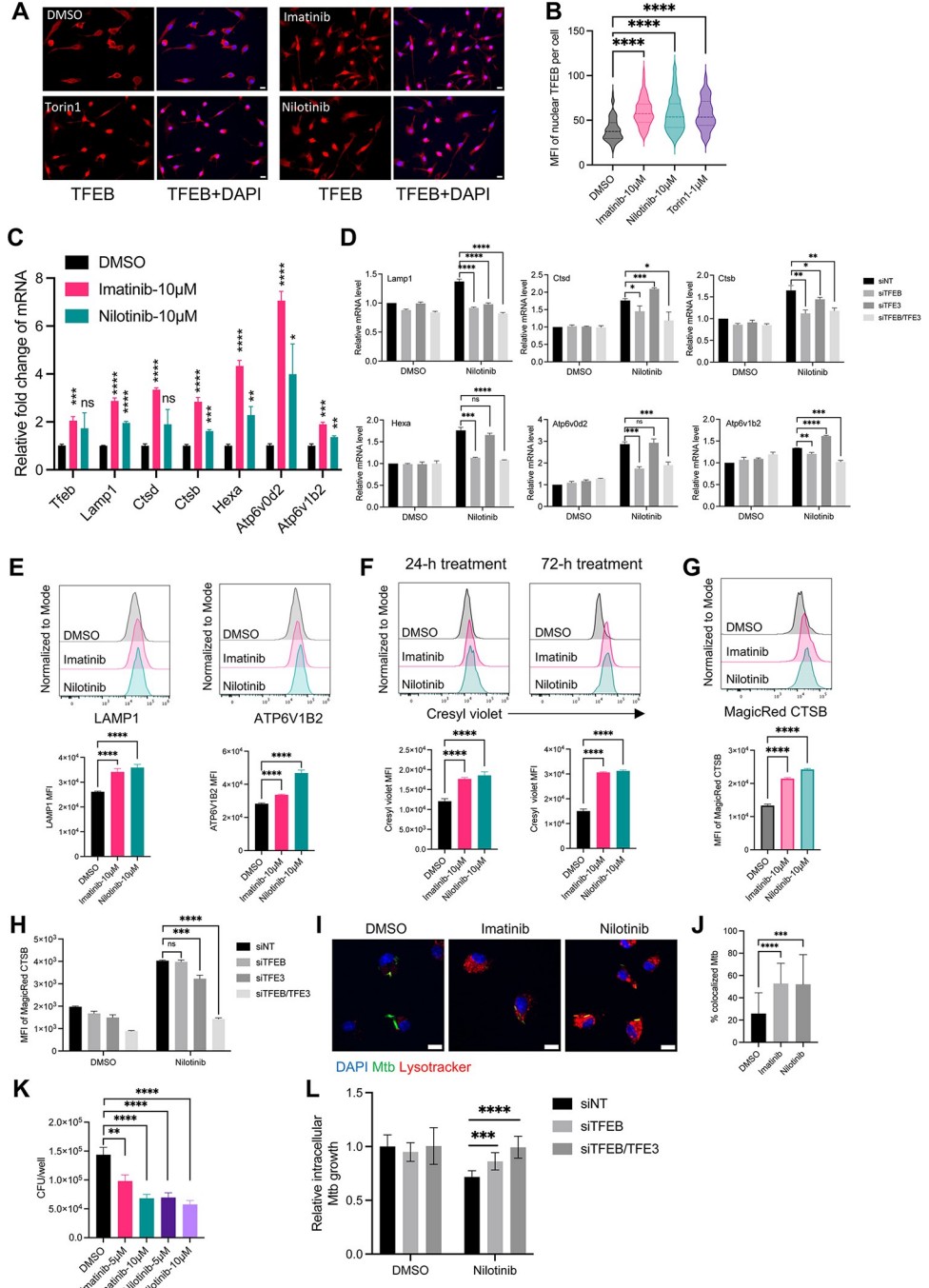

**Fig 7. c-Abl inhibitors activate TFEB, enhance lysosome biogenesis, and improve control of intracellular Mtb in cultured primary macrophages.** (A) Torin1 (mTOR inhibitor) and two c-Abl kinase inhibitors (imatinib and nilotinib) increase nuclear translocation of TFEB. BMDM were treated with DMSO or the indicated small molecules for 4 h. Then, cells were fixed and stained with an anti-TFEB antibody. Scale bar, 20 μm. (B) Quantification of nuclear TFEB MFI per cell from >57 cells for each condition in (A) using ImageJ. (C) Imatinib and nilotinib increase lysosome gene expression. qPCR results of lysosome genes for BMDM treated with DMSO or the indicated small molecules for 24 h (n = 3). (D) BMDM were transfected with the designated siRNA (20 nM) for 48 h, then treated with nilotinib (5 μM) for 18 h. The mRNA levels of lysosome genes were detected by qPCR. siNT: non-targeting siRNA control; n = 3. (E) Histograms and MFI of LAMP1 and ATP6V1B2 for uninfected BMDM treated with small molecules for 24 h. (F) Histograms and MFI of cresyl violet for uninfected BMDM treated with small molecules for 24 h or 72 h (n = 3). (G) Imatinib and nilotinib induce cathepsin B enzymatic activity. Histograms and MFI of fluorogenic cathepsin B product for H37Rv-infected BMDM treated with inhibitors for 72 h (n = 3). (H) MFI of fluorogenic

cathepsin B product in siRNA transfected BMDM (n = 3). (I) Imatinib and nilotinib enhance lysosome acidification. BMDM were infected with H37Rv-ZsGreen (MOI = 2), then treated with DMSO, imatinib (10 μM), or nilotinib (5 μM). BMDM were stained with DAPI and lysotracker at 24 hpi. Scale bar, 10 μm. (J) Imatinib and nilotinib enhance Mtb phagolysosome maturation in cultured primary macrophages. Quantification of Mtb co-localizing with lysotracker for (I) from >20 images per condition using JACoP in ImageJ. (K) Imatinib and nilotinib enhance control of Mtb in primary macrophages. BMDM were infected with H37Rv at MOI = 1 (n = 4), then treated with DMSO or indicated small molecules for 4 days. Cells were then lysed and plated for CFU assay. (L) siRNA-transfected BMDM were infected with Mtb H37Rv (MOI = 1) and then treated with nilotinib (2 μM) for 4d. Viable intracellular bacteria were quantitated by CFU assay. Shown are pooled data from two independent experiments (n = 8 for each condition). Results are presented as mean ± SD. Representative data from 2–3 independent experiments are shown for (A, B, F, G, K). *$p < 0.05$, **$p < 0.01$, ***$p < 0.001$, ****$p < 0.0001$ by unpaired Student's t test (B-D, E-H, J-L). ns: not significant.

and other cells (Fig 8C), treatment also led to a slight but not significant reduction of the frequency of Mtb+ cells in the MNC1 subset (S10B Fig). Consistent with the effects in cultured macrophages, nilotinib treatment enhanced the expression of multiple TFEB-activated genes in the lungs of infected mice (Fig 8D). Flow cytometry of lung cell subsets revealed that nilotinib treatment of Mtb-infected mice significantly increased protein expression of ATP6V1B2 in lung MNC1, MNC2, neutrophils and Ly6C$^{hi}$ monocytes (Fig 8E and 8F). Nilotinib treatment also enhanced lysosome acidification in MNC1, MNC2, neutrophils and Ly6C$^{hi}$ monocytes, but had a negligible effect on lysosome acidification in AM (Fig 8G). We then tested the effect of nilotinib treatment on lysosome CTSB activity in these lung cell subsets and found that nilotinib significantly increased CTSB activity in MNC1, MNC2 and Ly6c$^{hi}$ monocytes (Fig 8H). Unexpectedly, nilotinib was associated with slightly reduced CTSB activity in AM, which still maintained higher CTSB activity than other lung cell subsets. In addition, nilotinib reduced recruitment of MNC1, MNC2, neutrophils and Ly6C$^{hi}$ monocytes to the lungs (S10C Fig), consistent with a secondary effect on proinflammatory responses.

In addition, nilotinib treatment increased the frequency of activated (CD154$^{+}$) CD4 T cells in the lungs, while the overall frequencies of CD4$^{+}$ T cells, CD8$^{+}$ T cells, and CD8$^{+}$CD44$^{+}$ T cells were not different than in lungs of control mice (S10D–S10F Fig), implying that nilotinib-mediated increase of lysosome activity in antigen-presenting cells may promote development of antigen-specific CD4 T cells in vivo [57,58]. Collectively, these findings suggest that nilotinib rescues defective lysosomal functions of permissive myeloid subsets and improves the restriction of Mtb in the lungs in vivo.

## Discussion

We report here that CD11c$^{lo}$ monocyte-derived MNC1 cells are the most Mtb-permissive lung cell subset during chronic infection. Using two independent methods to quantitate live intracellular Mtb, we found that MNC1 are a major Mtb reservoir and importantly, harbor more live Mtb per infected cell than AM, neutrophils, or CD11c$^{hi}$ MNC2 at a stage of infection when adaptive immune responses have developed, and Mtb antigen-specific effector T cells are abundant in the lungs. These results extend our previous finding that monocyte-derived cells are the major infected populations during chronic infection [12]. Our results at 21 dpi are consistent with a recent report that CD11c$^{hi}$ monocyte-derived cells (MNC2 in this study) are the major infected subset at the same timepoint [8], yet they differ from those results at other time points, at least in part because we specifically quantitated live intracellular bacteria. Our results emphasize that Mtb infection is highly dynamic, especially before and during the development of adaptive immune responses, and that distinct lung myeloid cell subsets can prevail as niches for Mtb at different stages of infection.

Previous studies have shown that lysosomal enzymes are required for antimycobacterial activity [27–29]. In addition, monocyte-derived macrophages from TB resisters (TB contacts

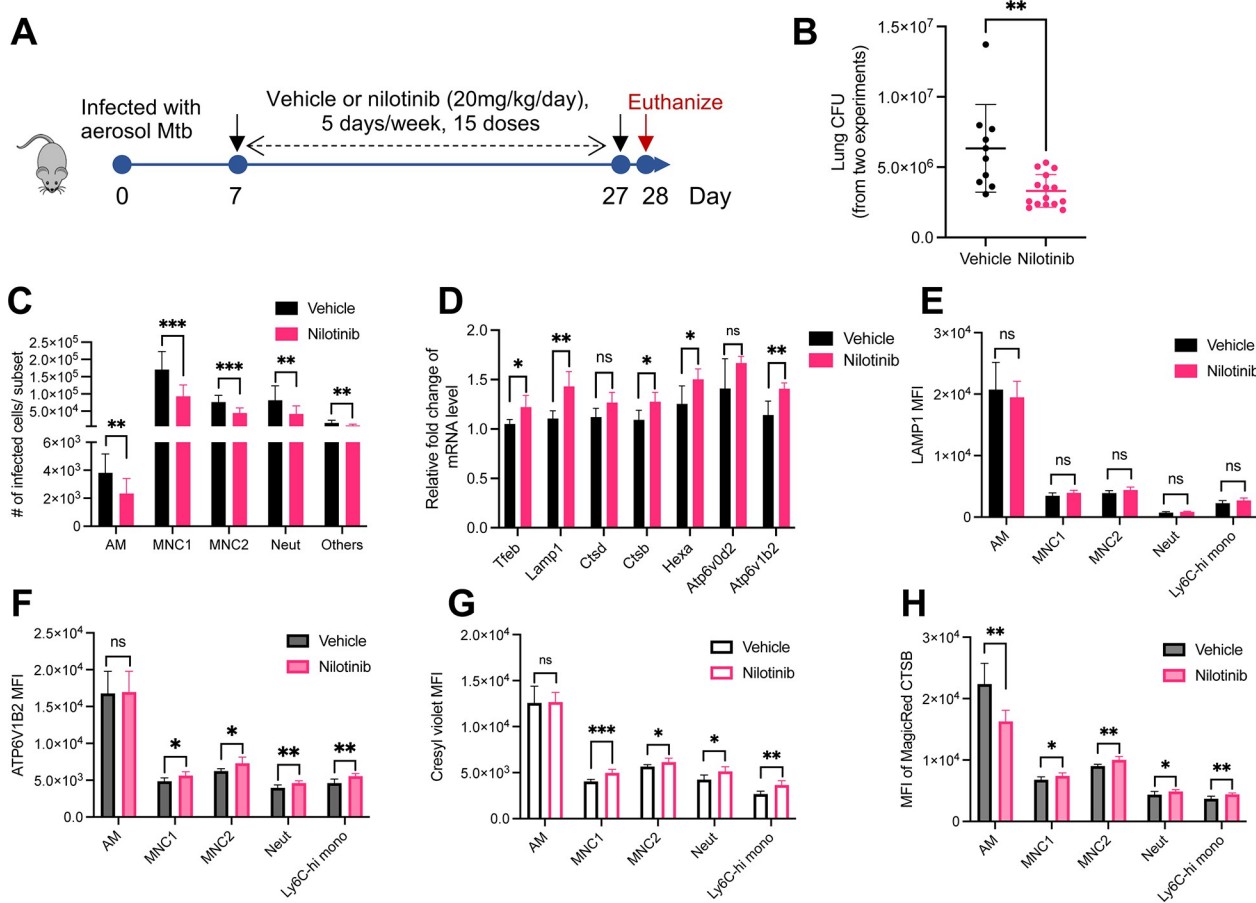

**Fig 8. Nilotinib treatment of Mtb-infected mice activates lysosome functions of lung myeloid subsets and reduces lung bacterial burdens.** (A) Experimental protocol for treatment of Mtb-infected C57BL/6 mice. Mice infected with Mtb H37Rv-ZsGreen were treated with vehicle or nilotinib (20 mg/kg/day) intraperitoneally, 5 days/week, for a total of 15 doses beginning at 7 dpi and ending at 27 dpi. At 28 dpi, mice were euthanized, and lungs were harvested for analysis. (B-C) Nilotinib treatment of mice (B) decreases overall Mtb lung bacterial burdens and (C) reduces infected lung myeloid cells in treated mice (28 dpi). Shown are pooled data from two independent experiments (Vehicle: n = 10; nilotinib, n = 15). (D) Nilotinib treatment of Mtb-infected mice increases expression of lysosome genes in lungs (qPCR analysis). Gapdh was used as a control (Vehicle: n = 4; nilotinib, n = 6). (E) Nilotinib treatment of Mtb-infected mice increases LAMP1 expression in recruited lung phagocyte subsets. Flow cytometry quantitation of intracellular LAMP1 MFI in lung subsets from mice infected with H37Rv-ZsGreen treated with vehicle or nilotinib (28 dpi, vehicle: n = 5; nilotinib, n = 8). (F) Nilotinib treatment of Mtb-infected mice increases lysosomal V-ATPase subunit ATP6V1B2 expression in recruited lung phagocyte subsets. Flow cytometry quantitation of intracellular ATP6V1B2 MFI in lung subsets from mice infected with H37Rv-ZsGreen treated with vehicle or nilotinib (28 dpi, vehicle: n = 5; nilotinib, n = 8). (G) Nilotinib treatment of Mtb-infected mice increases lysosomal acidification. Cresyl violet MFI quantitated by flow cytometry for lung subsets from H37Rv-ZsGreen-infected mice treated with vehicle or nilotinib (28 dpi, vehicle: n = 5; nilotinib, n = 8). (H) Nilotinib treatment of Mtb-infected mice increases lysosomal cathepsin B activity in recruited phagocytes, but not resident (alveolar) macrophages. MFI of fluorogenic MagicRed CTSB product in lung subsets from mice treated with vehicle or nilotinib (28 dpi) quantitated by flow cytometry (vehicle: n = 5; nilotinib, n = 8). Results are presented as mean ± SD. Representative data from two independent experiments are shown for (E, F, G). *$p < 0.05$, **$p < 0.01$, ***$p < 0.001$, by unpaired Student's t test (B-H). ns: not significant.

who developed neither asymptomatic LTBI nor active TB) exhibit better Mtb-killing ability via phagosome acidification compared to healthy controls, latent TB controls, and active TB patients [59]. We found that a monocyte-derived subset of cells, termed MNC1 are deficient in lysosome functions due to low lysosome content, poor lysosomal acidification and reduced enzyme activity compared to Mtb-restrictive AM, and lysosome functions in these lung cell subsets correspond to their ability to restrict intracellular Mtb (28 dpi). Lysosome deficiency of MNC1 is not due to a direct effect of Mtb or the essential virulence factor ESX-1 on individual cells. Instead, Mtb promotes the recruitment of permissive MNC1 and less permissive MNC2

to the lungs in an ESX-1 dependent manner. Therefore, in addition to blocking cell autonomous phagosome maturation and autophagy [2,60], Mtb exploits lysosome-poor monocyte-derived cells for persistence during chronic infection. Lysosome function is essential for autophagy, which is also required for restricting Mtb [61–65]. MNC1 may have a lower autophagy level than AM, which remains to be determined.

TFEB is a master transcription factor for lysosome biogenesis [41]. Compared with AM, MNC1 have less nuclear TFEB (i.e., activated TFEB) and lower expression of TFEB-activated genes at the mRNA and protein levels. This implies that TFEB activation regulates the difference in lysosome content and function between these subsets. The lysosome pathway was not differentially regulated between infected and uninfected cells for each lung cell subset in this study, suggesting that lysosomal genes may be regulated differently in different cell types. Notably, TFEB is required for IFNγ-dependent control of Mtb in cultured macrophages [45]. Other TFEB activators can boost the antimycobacterial activity of macrophages. These include bedaquiline [51], NR1D1 agonist [66], PPARα agonist [52], trehalose [67], and IFNγ [54]. We found that the c-Abl tyrosine kinase inhibitors imatinib and nilotinib activate TFEB and induce expression of downstream genes to improve lysosome function in macrophages. Like imatinib, nilotinib enhanced control of Mtb in vitro and in vivo, indicating that enhancement of lysosome function can partially overcome the reported inhibitory effects of Mtb on phagosome maturation. Interestingly, this activity is dependent on both TFEB and TFE3, suggesting a complex regulation of lysosome function and antimycobacterial activity by these transcription factors. In vivo, nilotinib improved lysosome functions in MNC1, MNC2, and Ly6C[hi] monocytes, suggesting the potential for enhancing lysosome functions in these Mtb-permissive monocyte-derived cells as a host-directed therapy for TB. TFEB-mediated lysosome biogenesis is also important for autophagy [41]. Like other TFEB activators, nilotinib has been reported to activate autophagy in macrophages [68]. Therefore, nilotinib and other TFEB activators likely improve Mtb restriction in part via enhancing phagosome maturation and autophagy. However, since kinase inhibitors including nilotinib can inhibit other kinases and have other activities in vivo, it is possible that other actions of nilotinib contribute to the in vivo antimycobacterial activity we observed.

Our study provides additional evidence that the ability of alveolar macrophages to permit or restrict Mtb evolves over the time of infection. AM serve as an early replication niche for Mtb [5,10], and are anti-inflammatory and permissive compared to IM in early innate immune responses to Mtb [7,10], due to induction of an NRF2-dependent cell-protective antioxidant response [10]. However, our results and others reveal that AM become activated and pro-inflammatory after the development of adaptive immune responses [8,23]. In line with this, AM upregulate lysosome function, which likely contributes to their better Mtb-restricting activity at later time points (after the development of T cell responses) [27–29,69]. Finally, the data generated using Mtb containing the live/dead reporter plasmid provide novel evidence that, at the time point studied here, AM can successfully kill virulent Mtb.

Although certain lysosome genes are highly expressed in neutrophils compared to MNC1, we did not find neutrophils have higher lysosome functions than MNC1, suggesting that different mechanisms regulate lysosome activities in neutrophils. Unlike mononuclear phagocytes, neutrophils do not have a classic endosomal pathway or classical lysosomes. Instead, they possess lysosome-like granules containing bactericidal factors that can rapidly fuse with phagosomes [70]. In addition, neutrophil phagosomes have neutral pH, and LAMP proteins are absent from their granules [70,71]. Thus, neutrophils use mechanisms distinct from those of mononuclear phagocytes to kill ingested pathogens, and these mechanisms (including NADPH oxidase activity) are not correlated with lysosome function.

In this work, the effect of nilotinib on lung bacterial burdens was modest. This may be due to Mtb inhibition of phagosome maturation even in the face of increased lysosome biogenesis; however, it is also possible that the limited effect is due to suboptimal pharmacokinetics or off-target effects that limit the positive effects.

A limitation of this study is the possibility that our cell subsets are heterogeneous, partially due to the lack of discriminating surface markers and limited fluorescence parameters of our BSL3-contained cell sorter. Although single-cell RNA sequencing has led to identifying other monocyte-derived cell subsets during Mtb infection [23], discriminating surface markers for better separation of these subsets still need improvement. Nevertheless, the resolution of our flow sorting strategy was sufficient to identify four transcriptionally distinct phagocyte subsets, including MNC1 that harbor the most live bacteria compared to the others. Further translational work will be required to ascertain lysosome function in lung mononuclear subsets in Mtb-infected humans; such studies face the challenges of obtaining lung parenchymal cells (and not only AM by bronchoalveolar lavage) from people with active tuberculosis.

In summary, our work revealed that Mtb recruits monocyte-derived MNC1 that are intrinsically deficient in lysosome biogenesis and that enable Mtb persistence during chronic infection (Fig 6E). The c-Abl inhibitor nilotinib activates TFEB and improves lysosome functions of monocyte-derived cells including MNC1, leading to enhanced control of Mtb in vitro and in vivo. Inhibiting c-Abl to enhance lysosome biogenesis represents a promising strategy for host-directed therapeutics to reprogram permissive cells to better control Mtb.

## Methods

### Ethics statement

All animal experiments were approved by the Institutional Animal Care and Use Committee of University of California, San Francisco or New York University. All mice were housed and maintained in specific pathogen-free conditions.

### Mice

C57BL/6 mice (8–12 weeks old) were obtained from Jackson Laboratory. Mice infected with Mtb were housed in Animal Biosafety Level 3 facility.

### Bacterial strains and growth

Mtb H37Rv was transformed with a plasmid (pMSP12::EGFP, pMV261::ZsGreen or pMSP12::mCherry) to constitutively express the respective fluorescent protein. H37Rv-live/dead was generated by transforming Mtb with the Live/Dead plasmid that drives constitutive expression of mCherry and tetracycline-inducible GFP [26]. BCG Pasteur was transformed with pMV261::ZsGreen. Bacteria were grown in Middlebrook 7H9 medium (BD) supplemented with 10% (v/v) ADC (albumin, dextrose, catalase), 0.05% Tween 80, 0.2% glycerol and 50 μg/ml kanamycin (for recombinant stains carrying pMV261 or pMSP12 plasmid).

### Chemicals and antibodies

All chemicals and antibodies for this study were listed in S1 Table.

### Generation and infection of BMDM

To generate BMDM [72], bone marrow cells were cultured in BMDM medium [DMEM (Gibco, 11965092), 10% heat-inactivated FBS (HI-FBS), and 20 ng/ml recombinant murine M-CSF (PeproTech)] for 6 days. Before infection, cells were washed twice with PBS and

harvested with PBS/2mM EDTA, resuspended in BMDM medium. $0.5 \times 10^5$ BMDM were seeded in 96-well plates. To prepare single-cell Mtb suspensions, 5mL mid-log Mtb cultures were centrifuged at 3000 g for 5min, washed in 5 mL DMEM/10% HI-FBS, and resuspended in 5mL BMDM infection media. The Mtb suspension was centrifuged at 234 g for 3min. 3mL of supernatant was transferred to a new tube and $OD_{600nm}$ was measured. Inoculum was prepared by diluting the supernatant with BMDM media. Media were aspirated and 100 μL of inoculum was added to the well. After incubation for 3h (37˚C, 5% $CO_2$), cells were washed three times with DMEM/1% HI-FBS and incubated in 200 μL of BMDM media containing small molecules of interest. After 4 days, medium was removed, and cells were lysed with 100 μL 0.1% Triton X-100 in sterile $H_2O$ for 5–10 min. Lysates were serially diluted in PBS/0.05% Tween 80, 50 μL of sample was plated on 7H11 agar plates. CFU were counted 3 weeks later.

### Aerosol infection

Mice were infected with Mtb or BCG via aerosol using an inhalation exposure unit from Glas-Col as previously described [9,12,18,21,22,72–83]. For controls of fluorescent protein expressing Mtb, mice were infected with H37Rv by the same procedure on the same day. Target dose was ~100 CFU/mouse for wild type Mtb, ~600 CFU/mouse for Mtb ΔRD1 strain, respectively. For BCG aerosol infection, we used a higher dose ~$2 \times 10^4$ CFU/mouse, as lower doses lead to bacterial clearance. Infection dose was determined by plating lung homogenates 24 hpi on 7H11 agar plates, and counting CFU after 3 weeks incubation at 37˚C.

H37Rv-live/dead carries a plasmid that drives constitutively expression of mCherry and tetracycline-inducible GFP. For induction of GFP expression in live bacteria, 1 mg/mL of doxycycline was given to mice via drinking water containing 5% sucrose for 6 days before day 28 harvest.

### Lung homogenate preparation

Lung homogenates were prepared as previously described with modifications [9]. Lungs were perfused with 10 mL of PBS/2 mM EDTA via right ventricle immediately after euthanasia. Lungs were processed with a gentleMACS dissociator (Miltenyi, lung program1), and digested in 4 mL of RPMI-1640/5% HI-FBS containing 1 mg/mL collagenase D (Sigma-Aldrich) and 30 μg/mL DNase I (Sigma) for 30 min at 37˚C. Digested lung tissues were further dissociated with the gentleMACS dissociator (lung program2) then passed through a 70-μm cell strainer. Red blood cells were lysed with 3mL ACK lysis buffer (Gibco) for 3 min and washed twice with RPMI-1640/5% HI-FBS. Lung cells were further processed as needed.

### Flow cytometry and cell sorting

Lung cells prepared as described above were counted and washed with cold PBS twice before staining with 1:200 Zombie Aqua Fixable Viability Dye (BioLegend, 423101) for 15 min at 4˚C, and then blocking with 1:100 CD16/CD32 (BD, 553142) for 10 minutes. Leaving the blocking antibody in, the cells were then stained with the antibodies diluted in Brilliant Stain Buffer (BD, 566349). Antibodies used for flow cytometry were listed in S1 Table. Surface staining was performed for 30 min at 4˚C, and samples were further processed as below.

For flow cytometry analysis, stained cells were washed twice with PBS and fixed overnight in 1% paraformaldehyde (PFA). Samples were analyzed using a BD LSRII or Sony MA900.

For live cell sorting, samples were washed twice with FACS buffer (PBS+2mM EDTA + 2% heat-inactivated FBS), resuspended in FACS buffer and passed through a 50 μm strainer

(concentration 10-20x10$^6$ cells/mL). Cell subsets were sorted using a Synergy cell sorter or Sony MA900 cell sorter through 100 μm nozzle.

For intracellular staining, cells were fixed and permeabilized using BD Fixation/Permeabilization Kit for 20 min at 4°C after surface staining, then washed and incubated with antibodies diluted in 1x BD Perm Wash/0.5% FBS for 30 min at room temperature. Samples were washed and acquired using a BD LSRII or Sony MA900.

## Bacterial quantitation in sorted cells

For CFU of infected cells, 1000 infected (mCherry$^+$ or EGFP$^+$) cells from each subset were sorted into RPMI 1640/5% HI-FBS, spun down and resuspended in 50 uL of 0.5% Tween 80/PBS. Serial dilutions were made using 0.05% Tween 80/PBS and plated on 24-well 7H11 agar plates. CFU were counted after 3 weeks. For quantitation of live and dead Mtb per cells, mCherry$^+$ cells were sorted into RPMI 1640/5% HI-FBS, spun down onto Shandon Cytoslide and fixed in 1% PFA overnight. Live (mCherry$^+$GFP$^+$) or dead (mCherry$^+$GFP$^-$) Mtb per infected cell ($\geq$ 300 cells/condition) were quantitated.

## RNA sequencing and data analysis

Ten thousand cells of each infected or bystander subset were sorted into RNAlater and stored at -20°C until use. Total RNA was extracted using RNeasy Plus Mini Kit (QIAGEN), and libraries were prepared by polyA selection using oligo-dT beads (Life Technologies) which were sequenced on the Illumina HiSeq 2500 following standard protocols to achieve 50 nucleotide, paired end reads.

RNA sequence processing and alignment: FastQC2 (v0.11.9) was used to generate quality-control reports of individual FASTQ files. Read were then aligned to the Mus musculus (house mouse) genome assembly GRCm38 (mm10) with the splice-aware STAR aligner using 'Gene-Counts' quant mode with the Ensembl gene transfer format file (indexes were built using '—sjdbOverhang 50' and the gtf file). Biotypes were investigated using PICARD tools (v2.22.3).

Differential gene expression and pathway analysis: Differential gene expression analyses were done in R using the DESeq2 package (v1.26.0) which uses a negative binomial generalized linear model. Blinded variance stabilizing transformation was applied and t-sne plots were then generated with Rtsne (v0.15) using PCA and a perplexity of three. Differentially expressed genes were identified using a Benjamini-Hochberg corrected alpha of 0.05 and an absolute effect size of one. Kyoto Encyclopedia of Genes and Genomes (KEGG; v4.0) pathway analyses were done using the R package clusterProfiler (v3.14.3). Analyses of custom pathways considering the gene background and not were done using camera and roast (using 99990 rotations) from the limma package (v3.42.2) respectively.

## Cathepsin enzymatic activity assays

Magic Red Cathepsin (B, L or K) assay kit (Bio-Rad) was used to determine the activity of cathepsin in cells. For immunofluorescence, subsets were sorted into RPMI1640/10% HI-FBS. 50,000 cells were plated in 8-well Nunc Lab-Tek chamber slides (Thermo Scientific, 177445), incubated in cell incubator for 1h. 200μL of media containing CTSB substrate (1:250 dilution) or CTSB+CTSL+CTSK substrates (1:750 dilution for each substrate) were added and incubated for 1h at 37°C. Control samples were treated with the substrates in the presence of bafilomycin A1 (100 nM).Cells were washed 2 times and fixed in 1% PFA overnight at 4°C. Images were taken using an in-house Leica fluorescent microscope or a CSU-22 Spinning Disk Confocal at UCSF Center for Advanced Microscopy.

For flow cytometry analysis, $1\text{-}2\times10^6$ lung cells or $0.5\times10^6$ BMDM in 24-well plates were incubated with CTSB substrate for 30 min at 37°C. Lung cells were washed twice in PBS and were processed according to the above protocol for flow cytometry. BMDM were washed and harvest with Cellstripper buffer. Data were acquired on a Sony MA900.

## Flow cytometric analysis of autophagy activity

Lung single cell suspensions were prepared as described above. $1\text{-}2\times10^6$ lung cells were incubated with 1x of the autophagy probe (ImmunoChemistry Technologies, 9156) in 200 μL RPMI/5%FBS for 30 min in the $CO_2$ incubator. Lung cells were washed twice in PBS and were processed according to the above protocol for flow cytometry.

## Staining of acidic lysosomes

Cresyl violet (Sigma) or LysoTracker DND-99 (Thermo Fisher) was used to label acidic lysosomes. For fluorescence analysis of sorted subsets or BMDM, cells were incubated with 5 μM cresyl violet or 200 nM LysoTracker DND-99 for 1 h at 37°C. Cells were washed 2 times and fixed in 1% PFA overnight at 4°C. Images were taken using an in-house Leica fluorescent microscope or a CSU-22 Spinning Disk Confocal. For flow cytometry, cells were incubated with 2 μM cresyl violet for 30 min at 37°C and processed as described in the cathepsin activity assays.

## Immunofluorescence

For immunofluorescence staining of BMDM or sorted cells, fixed cells were permeabilized with 0.1% Triton X-100/PBS for 10 min and blocked with 5% goat serum/PBS for 30min. Cells were then incubated with 1:500 primary antibody in 5% goat serum/PBS for 2 h at RT. Primary antibodies used were: anti-TFEB (Bethyl Laboratories, A303-673A), anti-TFE3 (Sigma, HPA023881), anti-LAMP1 (Invitrogen, 14-1071-82), anti-ATP6V0D2 (Sigma, SAB2103221, verified using isotype control), anti- ATP6V1B2 antibody (Santa Cruz, sc-166122, verified using isotype control) and anti-CTSB (Abcam, ab58802). Then cells were washed three times with PBS and stained with 1:1000 secondary antibody tagged with Alexa Fluor 647 (see S1 Table). Samples were washed three times with PBS and mounted ProLong Diamond Antifade Mountant with DAPI.

## Phagocytosis assay

Lung single-cell suspensions from Rv/ZsGreen-infected mice (28dpi) were harvested as described above. 1–2 million cells were incubated with Rv/mCherry (MOI = 5) for 1 h in the $CO_2$ incubator. After incubation, the cells were washed three times with 200 μL PBS and were further processed according to the above protocol of flow cytometry. Data shown were net MFI after subtracting the background MFI of the mCherry FMO control.

## Quantitative PCR (qPCR)

$1\times10^6$ BMDM were seeded in 6-well plates overnight, then treated with small molecules according to the specific experiment. After 24 h, the medium was removed, and cells were lysed with 350 μL TRIzol reagent (Invitrogen). Total RNA was extracted using a Direct-zol RNA Microprep kit with DNase I (R2062). 500 ng of RNA was used for cDNA synthesis using a PrimeScript RT Reagent Kit (Takara, RR037A). 2 μL of diluted cDNA (1:50) was used as the template for qPCR reaction using PowerUp SYBR Green Master Mix (Applied Biosystems, A25742). For internal control, Gapdh was used. Primers were listed in S2 Table.

### Fluorescent dextran feeding assay

Lung cells were incubated with 20 μg/mL (Invitrogen, D22914) of Dextran-Alexa Fluor 647 for 1h in cell incubator, washed once with RPMI/5%FBS, then incubated with RPMI/5%FBS for another 1h. Samples were then processed for flow cytometry analysis.

### siRNA transfection

siRNA transfection was performed using Lipofectamine RNAiMAX Transfection Reagent according to the manufacture's instruction. BMDM were reversely transfected with ON-TAR-GETplus SMARTpool siRNA (20 nM) targeting mouse TFEB (Dharmacon, L-050607-02-0005) or TFE3 (Dharmacon, L-054750-00-0005), and a non-targeting control siRNA pool (Dharmacon, D-001810-10-05). After 48 h, cells were used for experiments.

### Nilotinib treatment of Mtb-infected mice

Mice infected with Mtb H37Rv-ZsGreen were treated with vehicle or nilotinib (20 mg/kg/day) intraperitoneally, 5 days/week, for a total of 15 doses beginning at 7 dpi and ending at 27 dpi. At 28 dpi, mice were euthanized, and lungs were harvested for analysis. For preparation of 1mL nilotinib solution (4 mg/mL), nilotinib powder was completely dissolved in 100 μL of DMSO by sonication in an ultrasonic cleaner, then mixed well with 500 μL of PEG300, followed by the addition of 400 μL of sterile cell culture water. Vehicle solution was 10% DMSO + 50% PEG300 + 40% $H_2O$.

### Quantification and statistical analysis

Quantification of MFI from immunofluorescence images was done using ImageJ and data are background subtracted. MFI data shown for lung subsets were net MFI after subtracting the background MFI of the corresponding FMO control. Flow cytometry data were analyzed using FlowJo (version 10.8.1). GraphPad Prism software (version 9) was used for graphical presentation and statistical analysis. Results were presented as means ± SD. $*p < 0.05$, $**p < 0.01$, $***p < 0.001$, $****p < 0.0001$, ns = not significant. Statistical analysis and p values are stated in the figure legends and indicated in the figures respectively.

### Supporting information

**S1 Fig. Gating strategy used to identify myeloid subsets and infected cells.** (A) The representative flow panel to detect AM, neutrophil (Neut), MNC1, and MNC2 populations in lungs from mice infected with H37Rv-mCherry (28 dpi). After gating out B, T, and NK cells, AM are CD11b$^{lo}$CD11c$^{hi}$SiglecF$^{hi}$, MNC1 are SiglecF$^-$CD11b$^+$CD11c$^{lo}$MHCII$^+$, MNC2 are SiglecF$^-$CD11b$^+$CD11c$^{hi}$MHCII$^{hi}$, and neutrophils (Neut) are SiglecF$^-$Ly6G$^{hi}$CD11b$^{hi}$. (B) Illustrative plots of infected lung cells in each subset from mice infected with H37Rv-ZsGreen, H37Rv-GFP, or H37Rv-mCherry (28 dpi).
(TIF)

**S2 Fig. Use of MerTK and CD64 criteria excludes major fractions of recruited monocyte-derived cells and Mtb-infected cells during Mtb infection.** (A) Representative plots of MerTK and CD64 expression on lung subsets (grey dots), or infected cells (blue dots) from each subset of mice infected with H37Rv-ZsGreen (28 dpi). Lung subsets (AM, Neut, MNC1, and MNC2) were defined using gating strategy shown in Figure S1A, then were further analyzed for expression of MerTK and CD64. (B) MerTK$^+$CD64$^+$ frequency in each subset from mice infected with H37Rv-ZsGreen for 14–28 days of infection. (C) MerTK$^+$CD64$^+$ frequency

of infected cells in each infected subset from mice infected with H37Rv-ZsGreen for 14–28 days of infection. Results are presented as mean ± SD of 4–5 mice.
(TIF)

**S3 Fig. MNC1 harbor more bacteria at a later stage of chronic Mtb infection.** C57BL/6 mice were infected with low-dose aerosolized Mtb. At 28 dpi or 56 dpi, mouse lungs were harvested for flow cytometry analysis. (A) ZsGreen MFI of infected subsets from mice infected with H37Rv-ZsGreen (28 dpi). (B) MNC1 and MNC2 have a similar Mtb phagocytosis capacity, but lower than AM and neutrophils ex vivo. See Method detail. (C) Flow cytometry was used to analyze the subset population distribution of infected cells in mouse lungs infected with H37Rv-ZsGreen (56 dpi). (D) ZsGreen MFI of infected subsets from mice infected with H37Rv-ZsGreen (56 dpi). Results are presented as mean ± SD of 4–5 mice. *$p<0.05$, **$p<0.01$, ****$p<0.0001$ by one-way ANOVA.
(TIF)

**S4 Fig. Volcano plots showing differentially expressed genes between infected cells and bystander cells for each lung cell subset.** The green dot indicates significant genes with an adjusted p-value $\leq 0.05$ and a |log2 fold change| $\geq 1$, the red dot indicates non-significant genes, and the grey dot indicates genes filtered out of the analysis based on the Crooks index. The number of differentially expressed genes is 100, 7, 184 and 29 for Mtb-infected vs bystander AM, MNC1, MNC2 and Neutrophils, respectively.
(TIF)

**S5 Fig. Dot plot showing 18 KEGG pathways that differ significantly with an enrichment ratio greater than 0.04 for AM, MNC2, neutrophils, and MNC1.** The color represents the adjusted p values, the graph is ordered by descending values for MNC1 vs AM, while the dot size is proportional to the gene count.
(TIF)

**S6 Fig. MNC1 are deficient in lysosomal proteins at a later stage of chronic Mtb infection (56 dpi).** C57BL/6 mice were infected with low-dose aerosolized Mtb H37Rv-ZsGreen. Mouse lungs were harvested for flow cytometry analysis at 56 dpi. (A) LAMP1 MFI of lung subsets from H37Rv-ZsGreen-infected mice (56 dpi). (B) ATP6V1B2 MFI of lung subsets from H37Rv-ZsGreen-infected mice (56 dpi). Results are presented as mean ± SD of 4–5 mice. **$p<0.01$, ****$p<0.0001$ by one-way ANOVA ("not significant" was not shown).
(TIF)

**S7 Fig. Mtb ESX-1 promotes MNC1 recruitment.** C57BL/6 mice were infected with one of the three ZsGreen expressing strains via aerosol infection: Mtb H37Rv, Mtb H37Rv:△RD1, or *M. bovis* BCG. At 28 dpi, lungs were harvested for flow cytometry analysis or CFU assays. Naïve mice were uninfected. (A) LAMP1 MFI of lung subsets from naïve mice and infected mice (28dpi). (B) ATP6V1B2 MFI of lung subsets from naïve mice and infected mice (28dpi). (C) MFI of fluorogenic CTSB product for lung subsets from naïve mice and infected mice (28dpi). (D) Number of cells per subset from naïve mice, and mice infected with the indicated mycobacterial strains (28 dpi). (E) Subset fractions of total myeloid cells (28 dpi). (F) Frequency of cell types in total infected cells (28 dpi). (G) Lung CFU for mice infected with different mycobacterial strains (28 dpi). Results are presented as mean ± SD of 4–5 mice, representative of 2 independent experiments. **$p<0.01$ ****$p<0.0001$ by two-way ANOVA (A-F), or one-way ANOVA for (G). ("not significant" was not shown).
(TIF)

**S8 Fig. Effect of small molecule drugs with distinct targets on TFEB activation.** (A-B) BMDM were treated with indicated small molecules for 4h, then stained with DAPI and anti-TFEB for fluorescent microscopy. (C) Summary of the effect of small molecules on the activation of TFEB nuclear translocation. (D) Imatinib or Nilotinib do not inhibit H37Rv growth in 7H9 media. Three replicates per condition. Results are presented as mean ± SD, representative of 2 independent experiments.
(TIF)

**S9 Fig. c-Abl inhibitors imatinib and nilotinib induce TFE3 nuclear translocation in BMDM.** (A) BMDM were treated with indicated small molecules (Imatinib, 10 μM; Nilotinib, 10 μM) for 24h, then stained with DAPI and anti-TFE3 for fluorescent microscopy. (B) Quantification of nuclear TFE3 MFI per cell from >127 cells for each condition in (A) using ImageJ.
(TIF)

**S10 Fig. Effect of nilotinib treatment on mouse body weight, infection of lung myeloid cells, and lung T cells after Mtb infection.** Mice were infected with low-dose Mtb H37Rv-ZsGreen via aerosol, followed by treatment with vehicle or nilotinib (20mg/kg/day) intraperitoneally, 5 days/week, a total of 15 doses. Treatment was beginning on 7 dpi and ending on 27 dpi. Lungs were harvested for different assays at 28 dpi. (A) Fold change of mouse body weight on 28 dpi relative to that on 7 dpi. (B) Frequency of Mtb infected (ZsGreen+) cells in each subset (28 dpi). (C) Total number of cells in each lung subset from infected mice treated with vehicle or nilotinib (28 dpi). (D) Gating strategy for defining lung CD4$^+$ T cells (28 dpi). (E) Frequency of lung CD4$^+$ T cells, and frequency of CD154$^+$ cells in lung CD4$^+$ T cells (28 dpi). (F) Frequency of lung CD8$^+$ T cells, and frequency of CD44$^+$ cells in lung CD8$^+$ T cells (28 dpi). Results are presented as mean ± SD of 5–8 mice, representative of 2 independent experiments. $^*p<0.05$, $^{**}p<0.01$ by unpaired Student's t-test. ns: not significant.
(TIF)

**S1 Table. Reagents or Resource.**
(XLSX)

**S2 Table. Primers for qPCR analyses.**
(XLSX)

**S1 Data. GSE220147 raw counts.**
(CSV)

## Acknowledgments

We thank Dr. Jessica Jang for her contributions to initial experiments in this project, and Dr. Fei Ning for her assistance in mouse cell harvests. We also thank Dr. Samuel Behar for providing the live/dead reporter plasmid for Mtb.

## Author Contributions

**Conceptualization:** Weihao Zheng, Joel D. Ernst.

**Data curation:** Jason Limberis, Joel D. Ernst.

**Formal analysis:** Weihao Zheng, Jason Limberis, Joel D. Ernst.

**Funding acquisition:** Joel D. Ernst.

**Investigation:** Weihao Zheng, I-Chang Chang, Jonathan M. Budzik, Beth Shoshana Zha, Zachary Howard, Lucas Chen, Joel D. Ernst.

**Methodology:** Weihao Zheng, I-Chang Chang, Beth Shoshana Zha, Zachary Howard, Joel D. Ernst.

**Project administration:** Joel D. Ernst.

**Resources:** Jonathan M. Budzik, Joel D. Ernst.

**Supervision:** Joel D. Ernst.

**Validation:** Weihao Zheng, I-Chang Chang, Joel D. Ernst.

**Visualization:** I-Chang Chang, Jason Limberis, Joel D. Ernst.

**Writing – original draft:** Weihao Zheng, Joel D. Ernst.

**Writing – review & editing:** Weihao Zheng, Joel D. Ernst.

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
