## [Decision Letter · Decision Letter 0]

18 Feb 2024

Dear Prof. Ernst,

Thank you very much for submitting your manuscript "Mycobacterium tuberculosis resides in lysosome-poor monocyte-derived lung cells during chronic infection" for consideration at PLOS Pathogens. As with all papers reviewed by the journal, your manuscript was reviewed by members of the editorial board and by several independent reviewers. In light of the reviews (below this email), we would like to invite the resubmission of a significantly-revised version that takes into account the reviewers' comments.

It seems important to study some of the lysosome biology in careful detail including analysis of the MNC1 (vs MNC2) lysosomes with a protein expression-independent analysis. Including a comparison of MNC2 lysosomes as suggested by reviewer 2 in some of the key figures could address the differences in bacterial load in the different cell subsets. Answering the question as to the ability of AM to control replication of MTb would be an important contribution to the field. The finding that RNAseq analyses could not distinguish between the infection status of the cells seems important to evaluate. Finally, the question as to non-lysosomal effects of the inhibitors can likely be addressed.

We cannot make any decision about publication until we have seen the revised manuscript and your response to the reviewers' comments. Your revised manuscript is also likely to be sent to reviewers for further evaluation.

Sincerely,

Helena Ingrid Boshoff

Section Editor

PLOS Pathogens

Helena Boshoff

Section Editor

PLOS Pathogens

Michael Malim

Editor-in-Chief

PLOS Pathogens

orcid.org/0000-0002-7699-2064

It seems important to study some of the lysosome biology in careful detail including analysis of the MNC1 (vs MNC2) lysosomes with a protein expression-independent analysis. Including a comparison of MNC2 lysosomes as suggested by reviewer 2 in some of the key figures could address the differences in bacterial load in the different cell subsets. Answering the question as to the ability of AM to control replication of MTb would be an important contribution to the field. The finding that RNAseq analyses could not distinguish between the infection status of the cells seems important to evaluate. Finally, the question as to non-lysosomal effects of the inhibitors can likely be addressed.

Reviewer's Responses to Questions

**Part I - Summary**

Reviewer #1: Zheng et al., describe a population of monocyte-derived cells that form an important reservoir of Mtb infected cells during chronic infection. These monocyte-derived cells carry higher levels of bacteria and represent a highly permissive niche for Mtb replication, driven through the limited lysosome number observed in these cells. The authors further show that reduced lysosomal trafficking is due to lower levels of TFEB in this cell population and that pharmacologically enhancing lysosome levels through treatment with c-Abl inhibitors can reduce bacterial numbers. The experiments in the manuscript are very well done and nicely illustrate their point. The manuscript itself is engagingly written and reasoned. The findings will be highly interesting to both the tuberculosis community and the wider pathogenesis community as well as researchers working on lysosomal biogenesis in both infectious and non-infectious contexts. Overall, this is a quite strong manuscript with only small weaknesses to be addressed.

Reviewer #2: This study by Zheng et al. highlights a profound difference between two macrophage/monocyte populations in the lung, contrasting the resident alveolar macrophage and a particular recruited monocyte population (MNC1) with regards to their lysosome biology. Specifically, they find that, at later timepoints (Day 28), MNC1 harbor more viable bacteria than other myeloid subsets and that this can at least partially be attributed to their reduced lysosome levels and lysosomal function. They can boost lysosome biogenesis pharmacologically and show that this has significant, albeit modest, effects on reducing bacterial burdens across cell types. The studies are well executed and thorough with their investigation of this particular pathway.

Reviewer #3: This is an interesting study that look at lysosomal activity of monocytes recruited in the lungs of infected mice in the context of tuberculosis. There is a lot of novelty and interest in these findings as there has been very little in the literature regarding lysosomal activity in immune cell subsets in vitro and in vivo. The use of functional experiments to show lysosomal activity is highly appreciated, given that most studies will only show gene expression data. There are however several interpretations of the work and approaches that are not really supported by the data and in some cases incorrect

**Part II – Major Issues: Key Experiments Required for Acceptance**

Reviewer #1: A central intellectual thrust is the decrease in lysosome number in the MNC1 population. While the authors demonstrate reduced LAMP1 staining, as they note, this could be a product of either decreased lysosome number or decreased expression/intensity of LAMP1 per lysosome. It would further buttress their argument to use an expression-independent approach to enumerate lysosome number (e.g. feeding fluorescent dextran to the MNC1 population). While less critical, the authors might also consider performing additional staining with fluorescent dyes/sensors to get an approximate pH of the lysosomes within the MNC1 fraction compared to other fractions.

Reviewer #2: One major comment throughout the manuscript is that the authors heavily focus all comparisons on MNC1 vs AM. The data are abundantly clear and convincing that MNC1 show a multitude of lysosome defects as compared to AM, and yet the authors are also claiming that MNC1 are more permissive to Mtb than MNC2 without showing strong differences in lysosome biology between these two subsets. While this may be a difficult bar to achieve given that the two populations are clearly closely related, it begs the question as to why the viability of Mtb shown in Fig 1E still seems to be strongly different between the two monocyte populations, with only modest differences in lysosome biology shown throughout the paper (mostly in supplemental figures, ie Fig S6 and S7). It would be helpful if MNC2 were included in many of the early lysosome defect figures, such as Fig 3 and 4, alongside MNC1 and AM data. Perhaps the differences are small but persistent enough to explain how Mtb could still be controlled in the MNC2 subset. If the differences are not apparent, the authors should comment on other reasons there might be Mtb viability differences between MNC1 and MNC2. One possible difference of interest that the authors do not comment on is the different MHCII levels between the two subsets. While CD11c expression is primarily what is used to distinguish them, MNC2 seem to be quite a bit higher for MHCII MFI than MNC1. Can the authors comment on whether this might be impacting Mtb control in MNC2 on top of lysosome deficiencies? Perhaps MNC1 are more permissive because they do not interact with T cells as well as MNC2? This could also contribute to why this phenotype emerges at a later timepoint that coincides with T cell responses.

Given the precedence in the literature that AM are actually reservoirs for Mtb, it would be helpful if the authors could provide data using their live/dead reporter as to whether they corroborate or contrast these earlier findings. Are AM ineffective at killing Mtb at the earlier timepoints and there is a switch to Mtb killing at later timepoints? Or does your data suggest that they may always be more effective at eliminating bacteria than previously appreciated? This would be very useful information to clarify for the field.

It was surprising that the lung myeloid cells did not segregate by infection status in the RNAseq analysis given the published literature that infected and bystander cells do strongly segregate at earlier timepoints (Rothchild et al, up to 10 days post-infection, Pisu et al, 21 days post-infection). Do you think this is a timepoint-specific phenomenon where the transcriptional changes are shut off during chronic infection? Or that bystander cells experience so much antigen stimulation that they fully mimic infected cells at these later timepoints? It would be helpful to show whether you can replicate the published findings of clear transcriptional segregation between bystander and infected cells at earlier timepoints to ensure there isn’t a technical difference in the analyses and to highlight the unique nature of these chronic timepoints.

In the final figure, it is important to determine whether these inhibitors are ultimately affecting Mtb viability in MNC1 using the same live/dead reporter analysis shown in Figure 1. In addition, instead of plotting # infected cells/subset as in Fig 7C, can the authors plot this as % infected of each subset? That would help visualize how well the drug is impacting bacterial load in each type of immune cell. Finally, given the modest effects of nilotinib, can the authors comment on why this drug was chosen over imatinib? Maybe that drug would have performed better.

Reviewer #3: Some suggestions and recommendations below:

The use of a live dead reporter is very interesting. The system is based on doxycycline, a drug that eventually cannot reach all tissue environments. So, I would be more careful and consider the double positives active bacteria.

The concept of permissive is more related to replication, so to conclude on this aspect, it could be important to re-analyse the data and look at the total intensity per cell (e.g. as shown in Fig 1E)

In Figure 3G is not correct to suggest autophagy deficiency based only on this autophagy probe. The deficiency in proteolytic activity is clear and this will affect all lysosomal dependent degradation pathways. I suggest omitting the autophagy part here as it is irrelevant.

Antibodies against subunits for the vATPAse are notoriously bad (for staining in flow and immunofluorescence, tissues), so I suggest the authors be extra cautious here and add some controls that the antibodies work. The other experiments in Figure 4 (e.g. cresyl violet, Cathepsin substrates etc) are enough evidence for defective lysosomal function.

Nilotinib has many other effects that are lysosomal function independent. These results shown in Figures 6 and 7 are very interesting but I will be more cautions with the interpretation in the discussion.

**Part III – Minor Issues: Editorial and Data Presentation Modifications**

Reviewer #1: Line 75-78 - It’d be helpful to break up this sentence/streamline to make more clear

Line 324-329, line 414-416 – The lower recruitment of MNC1 and other cell types is considered as being evidence of RD1-dependent recruitment, however, given the lower burden in RD1 and BCG infected animals, these effects could also be secondary to the lower burden/decreased RD1-dependent growth. This should be discussed as an alternative possibility, as in the absence of matched infection burdens, it is hard to definitively establish that recruitment effects are directly dependent on RD1.

Figure 4 and S10 – Reorder the panels if possible.

Reviewer #2: It is appreciated that the authors have attempted to place their gating strategy into context with prior manuscripts and acknowledge heterogeneity in the MNC2 population, especially given the RNA-seq results that show the presence of DC as well as monocyte genes. Fig S2 is somewhat confusing as it attempts to compare the MNC1 and MNC2 as defined in this manuscript with prior studies using Mer-TK and CD64, but it is unclear how the cells in Fig S2 are pre-gated (if they are pre-gated?) and how this gating was subsequently applied to Fig S1 (as is mentioned in the figure legend). The figure shows that many cells fall outside the Mer-TK and CD64 gate, but among Mtb-infected cells it does seem to define the cells quite well, except for some CD64-negative cells in the MNC1 population. As such it would be nice to see additional flow cytometry using known markers to distinguish conventional DC from monocytes, including CD26, CD64, and MAR1, as in Bosteels et al, Immunity 2020. CD88 can also been used to distinguish monocyte-derived cells from DC.

What was the gating based on for MNC1 vs 2 for CD11c? Seems like the natural break in the populations is substantially lower.

Include negative control for representative flow cytometry plots in Fig S1 to determine how much fluorescent background might be present in each bacterial strain, especially given that the zsStrain seems to induce much higher levels of infection than either the mCherry or GFP strains.

Clarify if Fig 3B is showing MFI background-subtracted for the “no substrate” histograms to show change in fluorescence upon addition of substrate. While the difference between cell types is clear, they do have slightly different fluorescence signals at baseline that should be accounted for. Same comment for the autophagy figure (Figure 3H). In the latter, there appears to be much stronger induction of autophagy than cathepsin activity, but it is still lower in MNC1 than AM.

It would be nice to see representative flow plots for the staining in Fig 4C.

Page 8, line 248. There is a typo, with “different” needing to be deleted.

Did the authors use intravenous antibody labeling prior to euthanasia to differentiate monocytes in the parenchyma versus vasculature? There could be differences in such populations and it would be interesting if any of the

---

## [Editor Report · Decision Letter 1]

19 Apr 2024

Dear Prof. Ernst,

We are pleased to inform you that your manuscript 'Mycobacterium tuberculosis resides in lysosome-poor monocyte-derived lung cells during chronic infection' has been provisionally accepted for publication in PLOS Pathogens.

Best regards,

Helena Ingrid Boshoff

Section Editor

PLOS Pathogens

Helena Boshoff

Section Editor

PLOS Pathogens

Michael Malim

Editor-in-Chief

PLOS Pathogens

orcid.org/0000-0002-7699-2064

The authors have sufficiently addressed the reviewers' concerns.
---

## [Editor Report · Acceptance letter]

28 Apr 2024

Dear Prof. Ernst,

We are delighted to inform you that your manuscript, "Mycobacterium tuberculosis resides in lysosome-poor monocyte-derived lung cells during chronic infection," has been formally accepted for publication in PLOS Pathogens.

Best regards,

Michael Malim

Editor-in-Chief

PLOS Pathogens

orcid.org/0000-0002-7699-2064